# The mycobacterial glycoside hydrolase LamH enables capsular arabinomannan release and stimulates growth

Aaron Franklin [1], Vivian C. Salgueiro[2], Abigail J. Layton [1], Rudi Sullivan[1], Todd Mize[1], Lucía Vázquez-Iniesta[2], Samuel T. Benedict[1], Sudagar S. Gurcha[1], Itxaso Anso[3], Gurdyal S. Besra [1], Manuel Banzhaf[1], Andrew L. Lovering [1], Spencer J. Williams [4], Marcelo E. Guerin [5], Nichollas E. Scott [6], Rafael Prados-Rosales[2], Elisabeth C. Lowe [7] ✉ & Patrick J. Moynihan [1] ✉

Mycobacterial glycolipids are important cell envelope structures that drive host-pathogen interactions. Arguably, the most important are lipoarabinomannan (LAM) and its precursor, lipomannan (LM), which are trafficked from the bacterium to the host via unknown mechanisms. Arabinomannan is thought to be a capsular derivative of these molecules, lacking a lipid anchor. However, the mechanism by which this material is generated has yet to be elucidated. Here, we describe the identification of a glycoside hydrolase family 76 enzyme that we term LamH (Rv0365c in *Mycobacterium tuberculosis*) which specifically cleaves α−1,6-mannoside linkages within LM and LAM, driving its export to the capsule releasing its phosphatidyl-*myo*-inositol mannoside lipid anchor. Unexpectedly, we found that the catalytic activity of this enzyme is important for efficient exit from stationary phase cultures, potentially implicating arabinomannan as a signal for growth phase transition. Finally, we demonstrate that LamH is important for *M. tuberculosis* survival in macrophages.

The bacterial cell envelope plays multiple crucial roles, including providing cell shape, acting as a scaffold for various proteins, and shielding the cell from turgor pressure[1]. In mycobacteria, the fundamental constituents of the cell envelope include the mycolyl-arabinogalactan–peptidoglycan complex, alongside lipoglycans such as lipomannan (LM) and lipoarabinomannan (LAM) (Fig. 1a)[2]. Within the context of host–pathogen interactions, the chemical composition of these structures distinguishes the

bacteria from their host. This disparity facilitates host recognition of the pathogen while also allowing the pathogen to manipulate the host's immune response[2].

LM and LAM are important to mycobacterial pathogenesis, and LAM forms the basis of some diagnostic platforms[3]. They are required to control acidification of the phagosome and are a ligand for C-type lectins such as Dectin-2 and DC-Sign[4,5]. Through decades of careful analysis, the structure of LM and LAM has been revealed to include

[1]School of Biosciences, University of Birmingham, Birmingham, UK. [2]Department of Preventive Medicine, Public Health and Microbiology, School of Medicine, Universidad Autonoma de Madrid, Madrid, Spain. [3]Structural Glycobiology Laboratory, Department of Structural and Molecular Biology, Molecular Biology Institute of Barcelona, Spanish National Research Council, Barcelona Science Park, c/Baldiri Reixac 10-12, Tower R, 08028 Barcelona, Catalonia, Spain. [4]School of Chemistry and Bio21 Molecular Science and Biotechnology Institute, University of Melbourne, Parkville, VIC, Australia. [5]Structural Glycobiology Laboratory, Department of Structural and Molecular Biology; Molecular Biology Institute of Barcelona (IBMB), Spanish National Research Council (CSIC), Barcelona, Catalonia, Spain. [6]Department of Microbiology and Immunology, University of Melbourne at the Peter Doherty Institute for Infection and Immunity, Melbourne, VIC, Australia. [7]Newcastle University Biosciences Institute, Medical School, Newcastle University, Newcastle upon Tyne, UK. ✉e-mail: elisabeth.lowe@ncl.ac.uk; p.j.moynihan@bham.ac.uk

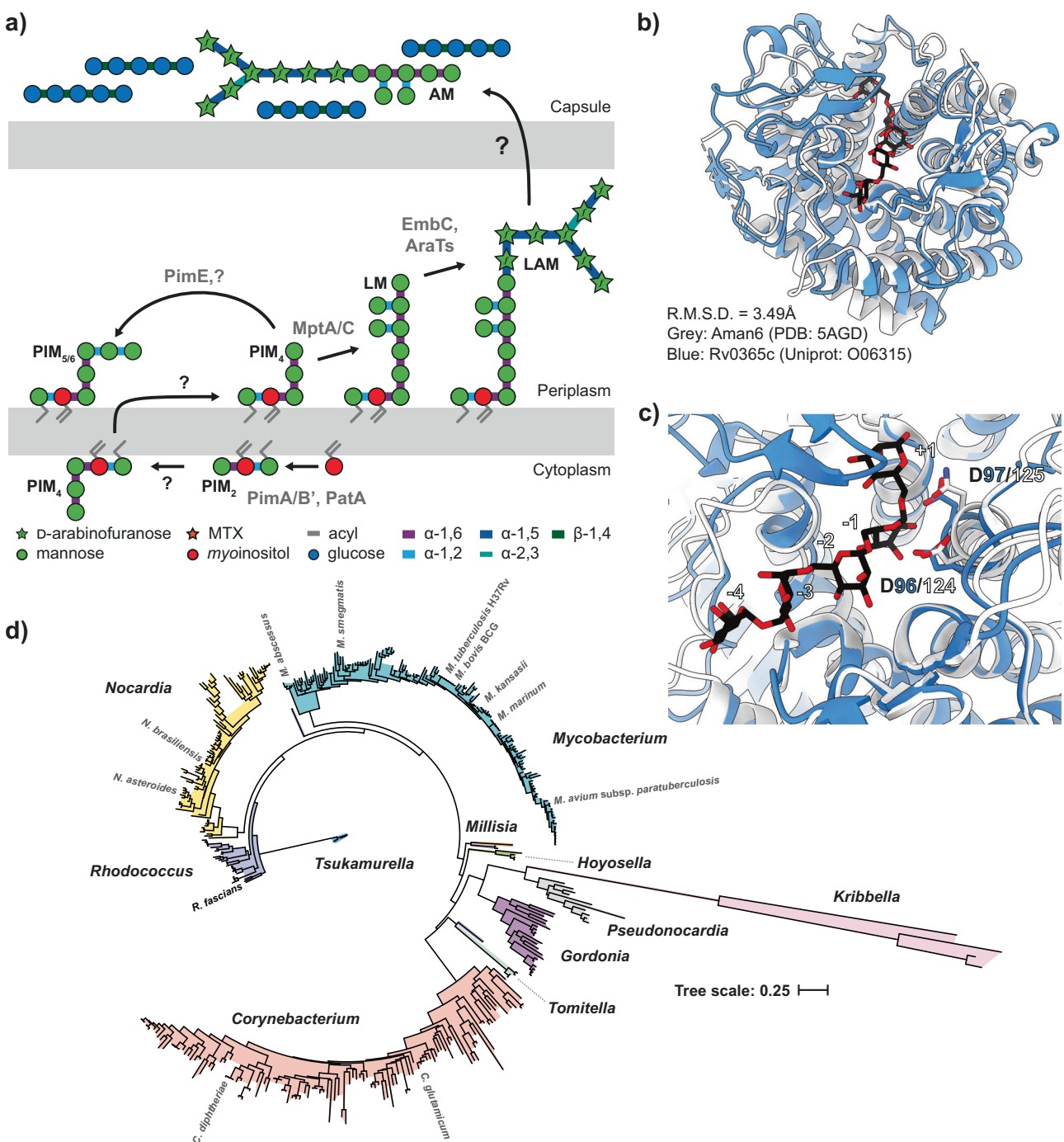

**Fig. 1 | Biosynthesis of mannosylated glycolipids in mycobacteria. a** The biosynthetic pathway for LM and LAM results in three primary products: PIMs, LM and LAM. The degree of acylation for each glycolipid can vary, as can secondary modifications on LM and LAM. An unknown process drives the expression of AM and M on the surface of the bacteria as part of their capsule. A simplified structure of LM/LAM is presented for clarity. The grey text indicates the presumed enzyme catalysing the reaction, and the black text indicates the glycolipid species. Several of the enzymes and the exact location for some steps in this process remain unknown. **b** Ribbon diagram of Aman6 (PDB:5AGD; grey) in complex with α-1,6-mannotetraose aligned in ccp4i Superpose with the AlphaFold 2.0 predicted structure for LamH (Rv0365c) associated with its Uniprot entry (O06315; blue). **c** The confirmed catalytic residues in Aman6 (D124/D125; D125N mutant in 5AGD; grey) are conserved in LamH (D96/D97; blue). **d** Phylogenetic tree of GH76 enzymes from the Mycobacteriales. Genomes for all available members of the Mycobacteriales were used to generate a custom BLAST database in Geneious Prime 2023.1.1. LamH was used as a query in a BLAST search of this database, yielding high-confidence homologues from all species. This list of proteins was submitted to NGPhylogeny.fr using the PHYML/OneClick tool; the tree was then manually coloured to identify individual genera[37]. Source data are provided as a Source Data file.

three major structural domains. The first is a phosphatidyl-*myo*-inositol anchor, believed to be biosynthetically derived from phosphatidyl-*myo*-inositol mannosides (PIMs)[6]. Attached to this is a second major domain, a large mannan, comprised of an α-1,6 mannose backbone of approximately 13 residues[7]. This is further branched with minor α-1,2

mannose decorations[7]. The final principal structural element is a large, branched, D-arabinofuranose domain that is far more complex and includes minor modifications such as succinyl groups[8–10]. Much of the immunogenicity of LAM is associated with the capping structures attached to the terminal β-linked D-arabinofuranose moieties of the D-

arabinan domain. This is most notable in the form of α-1,2-linked mannose caps and an unusual methylthioxylose residue in *M. tuberculosis*[11–14]. While broadly similar across species, the precise structure of LM and LAM can vary between strains, substantially impacting the host response[15]. Moreover, although chemical analysis of bacterial-derived LAM has been valuable, the structures of the macromolecules ultimately secreted and recognised by the host are not well-defined. LM and LAM are also suggested to contribute to the bacteria's fundamental biology, aiding proper septal formation analogous to lipoteichoic acids in some Gram-positive bacteria[16,17]. Nonetheless, much of the biology of LM and LAM remains to be uncovered.

Amongst the pathways that require further study are the modification and release of LM and LAM. Release of these molecules may happen during cell wall degradation because of division and wall remodelling or as an apparent virulence factor as occurs for peptidoglycan in *Bordetella pertussis*, and *Neisseria gonorrhoeae*[18–20]. In the context of mycobacteria, we recently described the identification of a family of glycoside hydrolases (GHs) responsible for the release of D-arabinan fragments into the surrounding environment[21]. However, the fate of these fragments is largely unknown, although at least some of these molecules are found within the mycobacterial capsule[22]. The composition of this capsule varies substantially amongst mycobacterial species, but, in *M. tuberculosis*, the carbohydrate component consists of 80% α-glucan and 20% arabinomannan (AM) and mannan[22]. AM and mannan are believed to be derived from LM and LAM, although the mechanism for this remains to be established.

To address this gap in our understanding, we have identified LamH (Rv0365c in *M. tuberculosis*) as the enzyme responsible for releasing the carbohydrate domain of LM and LAM into the capsule of mycobacteria. We show that LamH is a glycoside hydrolase that is specific for the attachment point between the lipid anchor and the carbohydrate domain of LM and LAM, highlighting its role in the biology of the bacilli. Loss of LamH extends the lag phase and is coupled with an accumulation of LM and LAM and a down-regulation in the production of each of these molecules. In addition, our data show that capsular AM derived from LamH-action AM aids the transition to exponential growth. Finally, we show that this protein facilitates the correct processing and display of capsular AM and mannan and that the knockdown of LamH in *M. tuberculosis* decreases bacterial fitness in macrophages.

## Results

### Mycobacteria encode a single predicted family GH76 enzyme

Previous studies have identified the GH76 family as a large group of enzymes capable of degrading or, in a subset, generating α-1,6-mannoside linkages through transglycosylation[23–26]. For example, we previously demonstrated that GH76 enzymes from the Bacteroidota are essential for the cleavage of α-1,6-mannoside linkages in fungal mannan in the human gut[25]. Using the experimental structure of Aman6 (PDB:5AGD), a known GH76 family member, as a search model in Foldseek, we limited the search taxon to *M. tuberculosis* H37Rv[27]. This search yielded a single high-confidence result, supporting the identification of Rv0365c as the sole predicted family GH76 enzyme encoded within the *M. tuberculosis* genome. While structurally similar to Aman6 (LSQKab calculated R.M.S.D. = 3.49 Å), the mycobacterial enzyme is predicted to possess an additional β-hairpin cap covering the active site (Fig. 1b, c)[28]. Proteomics studies have localised Rv0365c to the cytoplasmic membrane or the cell wall fraction in *M. tuberculosis* and *Mycobacterium smegmatis* (MSMEG_0740)[29–33]. There is no detectable signal sequence on Rv0365c, however there is precedent in the literature for mycobacterial proteins secreted without an identifiable signal peptide or transmembrane helix[34–36]. Subsequently, we investigated the conservation of Rv0365c homologs among mycobacteria. Using a custom BLAST database comprising representative

genomes of Mycobacteriales, we identified Rv0365c homologs and NGPhylogeny.fr to reconstruct a phylogenetic tree (Fig. 1d)[37,38]. Rv0365c homologs were conserved in all representative species, suggesting their involvement in an evolutionarily conserved process.

### Rv0365c specifically cleaves α-1,6-mannoside linkages

Next, we wanted to determine if Rv0365c possessed GH76-like catalytic activity. LM and LAM are complex substrates, making them unsuited for determining the precise linkage specificity of the enzyme. To address this, we isolated mannan from three *Saccharomyces cerevisiae* strains (Mnn1, Mnn2 and Mnn5) that lack mannan glycosyltransferases and thus produced three distinct mannans of decreasing complexity (Supplementary Fig. 1a)[25]. The mannan derived from *S. cerevisiae* Mnn2 comprises a backbone of α-1,6-linked mannose, while mannan from *S. cerevisiae* Mnn5 and Mnn1 have additional α-1,2-linked mannose decorations and extensions, respectively[24]. Initial assays with purified Rv0365c on these substrates yielded no observable reaction products (Fig. 2a). Many endo-acting GHs exhibit a preference for shorter substrates in vitro. Considering this possibility, we pre-digested the yeast mannans with BT3792, a GH76 family member from *Bacteroides thetaiotamicron* previously shown to produce a mixture of α-1,6 mannan oligosaccharides from fungal mannan[24]. Incubation of these pre-processed mannans with Rv0365c generated oligosaccharide products consistent with endo-activity (Fig. 2b). Notably, the enzyme only digested material derived from the *S. cerevisiae* strain Mnn2, which lacks α-1,2-linked mannose decorations (Fig. 2b). These data demonstrate α-1,6-mannanase activity for Rv0365c and indicate that the enzyme is unable to cleave substrates with α-1,2-linked mannose decorations. To confirm this finding, we isolated an α-1,6-mannotetraose oligosaccharide from the BT3792 digestion of Mnn2 mannan (Supplementary Fig. 2a, b). Rv0365c displayed activity on this substrate, producing mixed mono- and oligosaccharide products and supporting its designation as an endo-α-1,6-mannanase (Fig. 2c). We also incubated the enzyme with capsular α-glucan to determine if it could process this polysaccharide. Despite prolonged incubation (18 h) under conditions for which the enzyme is active on mannan, no product formation was observed (Supplementary Fig. 2c). Taken together, these results show that Rv0365c is a member of the GH76 family and is active on undecorated α-1,6 mannan.

**Rv0365c cleaves the lipid anchor from LM and LAM**. In mycobacteria, α-1,6-linked mannan has only been identified in regions of LM, LAM, and their presumed capsular derivatives. We, therefore, hypothesised that Rv0365c would specifically degrade LM and LAM. To test this hypothesis, we purified these lipoglycans from *Mycobacterium bovis* BCG Danish 1331 and incubated them with the enzyme. Analysis of the reaction products by TLC, using a solvent system that retains the carbohydrate domain at the origin and separates any released glycolipids reveals that Rv0365c released a low molecular weight product from LM and LAM, which stained positive for carbohydrates and lipids (Fig. 2d). In addition, we separated the LM/LAM reaction products by SDS-PAGE followed by Pro-Q Emerald staining, a reagent that reacts with some glycans to produce an easily visualised fluorescent product. These experiments demonstrated enzyme activity on both LM and LAM (Fig. 2e). Conversely, incubation of the enzyme with isolated PIMs yielded no detectable products, suggesting it cannot process $PIM_5$ or $PIM_6$ substrates, consistent with its inability to degrade substrates with α-1,2-linked mannose decorations (Supplementary Fig. 1b).

Drawing from the comparison to Aman6 (Fig. 1c), we predicted that D96 and D97 serve as the catalytic acid/base and nucleophile residues of LamH. This conjecture is supported by the lack of activity for the D96A mutant (Fig. 2e). Next, we sought to determine if Rv0365c exhibited a preference for LM or LAM. Quantification of the

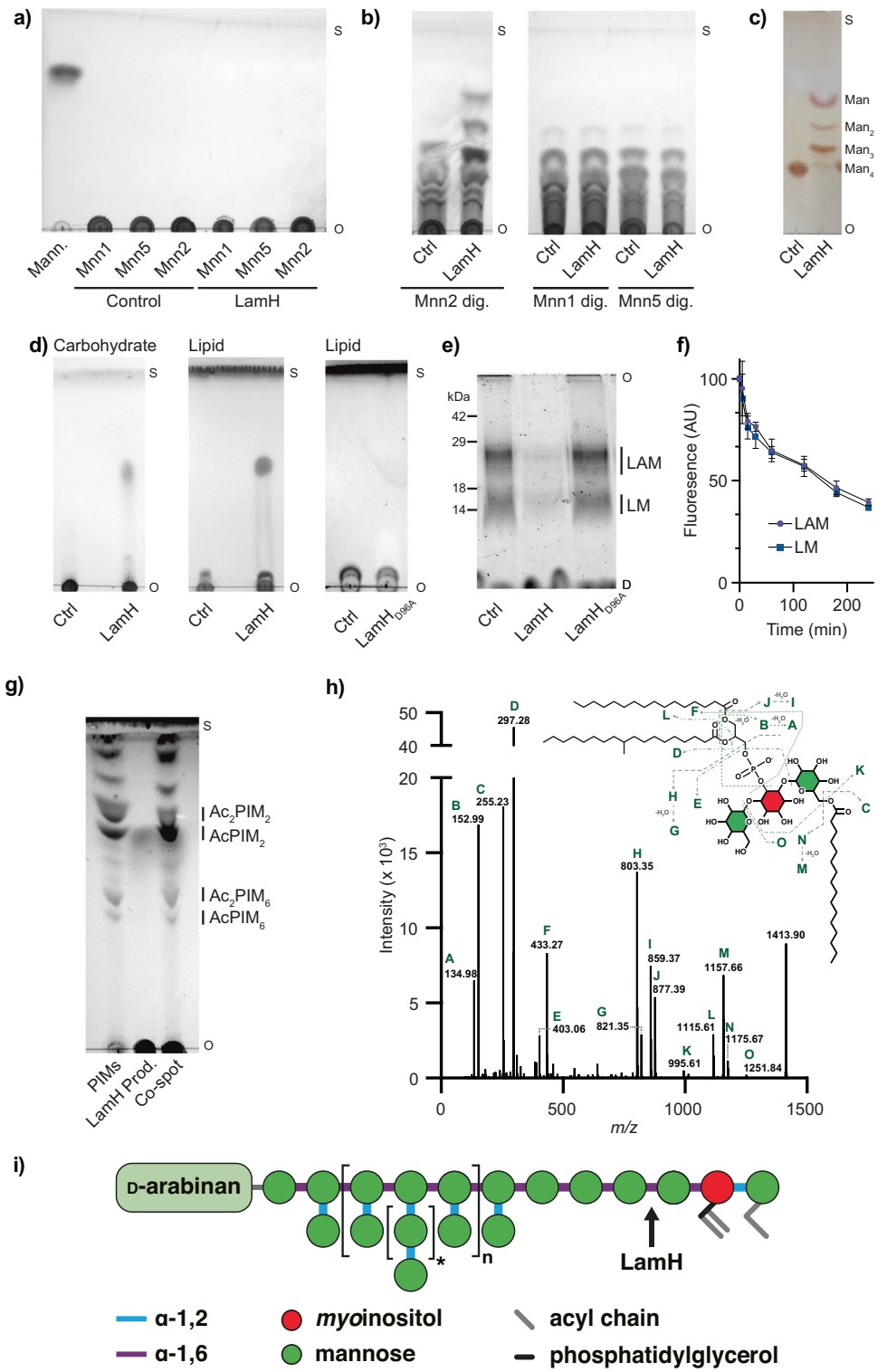

degradation of both species over time, by incubating the enzyme with LM/LAM and then analysing a time course of the reaction products by SDS-PAGE (Supplementary Fig. 1c), revealed that the LM and LAM bands decrease at equal rates (Fig. 2f), indicating no substrate preference. Prior work suggests that there are, on average, between 2 (*M. tuberculosis*) and 7 (*M. smegmatis*) α-1,6-linked mannose residues directly attached to the *myo*-inositol anchor, which lack α-1,2 decorations[8,39]. In this context, given the many possible cleavage sites available for Rv0365c, we then aimed to identify the glycolipid product

formed by Rv0365c digestion of LAM. Analysis of this glycolipid product by MS/MS revealed it to be AcPIM$_2$ (Fig. 2h and Supplementary Table 1). This assignment is supported by TLC analysis, where the product co-migrates with AcPIM$_2$ (Fig. 2g). The cleavage site on LAM is illustrated in Fig. 2i, using the most recent structural proposal of LAM from *M. tuberculosis*[7]. Based on these data, we conclude that Rv0365c degrades LM and LAM, releasing the carbohydrate domain from the AcPIM$_2$ lipid anchor and consequently have renamed the protein LamH (Lipoarabinomannan Hydrolase).

**Fig. 2 | LamH (Rv0365c) is a GH76 family enzyme that cleaves LM and LAM to produce AcPIM₂.** Yeast mannan substrates derived from the indicated strains were incubated with LamH for 16 h at 37 °C without (**a**) or with (**b**) pre-digestion by BT3792. Products were separated by TLC (2:1:1, butanol:acetic acid: water) and visualised by staining with orcinol. **c** LamH was incubated with purified α−1,6-mannotetraose, and the reaction products were analysed by TLC as above. **d** LamH or LamH$_{D96A}$ was incubated with mixed LM/LAM, and the reaction product was analysed by TLC (chloroform:methanol:13 M ammonia:1 M ammonium acetate: water (180:140:9:9:23 v/v/v/v/v)). Duplicate TLCs were stained with either orcinol or phosphomolybdic acid, indicating the presence of carbohydrates and lipids, respectively. **e** An aliquot of the reaction products analysed in (**d**) was separated by SDS-PAGE, and glycolipids were detected using Pro-Q Emerald staining. **f** LamH activity against LM and LAM was compared by separating reaction products at the indicated time points and quantifying the LM and LAM fluorescence in Pro-Q Emerald stained SDS-PAGE gels. Error bars represent the standard deviation of 3 biological replicates with the centre of measure defined as the average. **g** The glycolipid product identified in (**d**) was separated by TLC adjacent to it and co-spotted with a preparation of PIMs from *M. bovis* BCG. **h** The small glycolipid identified in (**b**) was isolated and analysed by MS/MS. The fragmentation pattern is consistent with an AcPIM₂ species. A table of identified peaks is presented in Supplementary Table 1. **i** Schematic diagram of LM/LAM based on ref. 7 with the site of LamH activity indicated. All TLCs or gels are representative of 3 biological replicates. O origin, S solvent front, D dye front. Source data are provided as a Source Data file.

## LamH drives the production of capsular (arabino)mannan

Our biochemical findings provide evidence supporting the hypothesis that LamH is responsible for generating LM and LAM capsular products. To assess this, we utilised a transposon mutant within *lamH* (BCGDAN_0378) in *M. bovis* BCG Danish 1331 (*lamH::Himar1*), subsequently referred to as Δ*lamH*[40]. This species shares 99% genetic similarity with *M. tuberculosis*, having primarily lost elements related to pathogenesis, and serves as a widely accepted model system for *M. tuberculosis* envelope biogenesis and turnover[41]. Initially, we aimed to determine whether loss of *lamH* affected LM and LAM levels in the mutant. At mid-exponential growth, we observed approximately 30% more LM/LAM in the Δ*lamH* strain compared to the wild-type strain (Fig. 3a, b and Supplementary Fig. 3a). Consistent with the biochemical data indicating no preference for cleavage of LM and LAM by LamH, the increase in accumulation was approximately equal for both LM and LAM. Subsequently, we generated a complementation vector in the pMV306 plasmid, which lacks a constitutive promoter. This construct contained the *lamH* open reading frame and 300 upstream bases to include possible promoter elements. Introduction of this *lamH* locus at the attP recognition site of mycobacteriophage L5 site fully restored the normal levels of LM/LAM in the complemented mutant, while a catalytically inactive variant of this construct phenocopied the Δ*lamH* strain (Fig. 3a, b)[42]. In addition, we deleted *lamH* (MSMEG_0740) in *M. smegmatis* mc²155 using ORBIT-mediated mutagenesis and analysed this strain's LM/LAM composition, giving results consistent with those observed in *M. bovis* BCG Danish (Supplementary Fig. 4a, b)[43]. Collectively, these data provide further support for the hypothesis that LamH regulates LM/LAM levels within the cell.

To investigate if LamH is responsible for the production of capsular (arabino)mannan, we measured levels of AM in the capsule of wild-type and mutant strains. Capsular polysaccharides from these strains were isolated from bacteria cultivated on solid media and labelled with 2-aminobenzamide (2-AB). Subsequently, these were separated by size-exclusion chromatography, allowing separation of α-glucan (~100 kDa) from AM (~14 kDa)[22]. As shown in Fig. 3c, while AM was detected in the wild type, none was detected in the capsule of the Δ*lamH* strain, with capsular AM production being restored in the complemented strain. Moreover, catalytic activity of the enzyme was found to be necessary for the production of capsular AM (Fig. 3c). As 2-AB labelling instals a single label at each reducing end of the glycans, it allows calculation of the ratio of reducing ends of α-glucan to AM, serving as a proxy for capsular composition. This analysis indicated that the complemented mutant produced significantly less AM than the wild-type (Fig. 3d), suggesting imperfect complementation and potential alteration in *lamH* gene expression when located distally. We also analysed the PIM composition of the wild-type and mutant bacteria (Fig. 3e and Supplementary Fig. 3b). These data indicate that in response to *lamH* deletion, the bacteria produce less AcPIM₂ but significantly more Ac₂PIM₂. A similar, though less pronounced, trend was observed in the *M. smegmatis* mc²155 Δ*lamH* strain (Supplementary Fig. 4c). Recent findings have shown that acylation of PIMs can occur as

a response to membrane stress[44]. In this context, the increased abundance of LM/LAM may induce a membrane stress response, resulting in increased levels of Ac₂PIM₂. Taken together, these data are consistent with the hypothesis that LamH drives the production of capsular AM by cleaving LAM.

## Mycobacteria respond to the lack of LamH activity by down-regulating LAM biogenesis

Given that Δ*lamH* bacteria were unable to degrade existing LM/LAM, it was surprising that they did not accumulate even more of these glycolipids. To gain insight into this phenomenon, we conducted whole-cell proteomics of mid-exponential bacteria (Fig. 4a, b, Supplementary Fig. 4 and Supplementary Data 1). Applying a threshold of $P < 0.01$ and > ±1-fold change, we observe 215 proteins decrease in abundance and 29 increased in abundance in response to the loss of *lamH*. The Δ*lamH* strain showed a marked reduction in the abundance of several enzymes involved in the LM/LAM biogenesis pathway, indicating a bacterial response to the accumulation of LM/LAM by decreasing synthesis (Fig. 4a, b). While not all biosynthetic enzymes were observed, even within the wild type, levels of arabinosyltransferases associated with LAM biogenesis (AftB, AftC, AftD) were significantly reduced or undetectable within Δ*lamH* (Supplementary Fig. 5)[45–48]. Similarly, levels of EmbC and MptA, which are arabinosyl- and mannosyl-transferases associated with LAM biogenesis, showed a reduction albeit not statistically significantly so[49,50]. In contrast, the abundance of PimA and PatA, which are involved in PIM and LM/LAM biogenesis, was unchanged (Supplementary Fig. 5)[51–54]. No significant changes were observed for galactan synthases such as Glft1 and 2 or key mycolic acid biogenesis proteins such as the antigen 85 complex, Mmpl3 and Pks13.

To identify functionally clustered groups of proteins with varying abundance, we analysed the dataset using Gene Ontology (GO) enrichment analysis (Supplementary Data 1b and Supplementary Fig. 6). These results revealed an over-representation of proteins decreased more than twofold in the observable proteome of the Δ*lamH* strain associated with the GO terms 'plasma membrane (GO:005886)' and 'peptidoglycan-based cell wall (GO:0009275)'. Conversely, no proteins were observed that were differentially regulated more than 2-fold associated with 'translation (GO:0006412)' and 'structural constituent of the ribosome (GO:0003735)'. These data provide evidence of bacterial regulation of LM/LAM levels implicating LamH in maintaining LM/LAM homoeostasis.

## LamH is required for an efficient transition from lag-phase growth

Given that LAM was recently reported to be involved in septation, we sought to investigate whether the absence of LAM turnover affects growth kinetics[55]. Surprisingly, in *M. bovis* BCG Danish 1331, loss of LamH prolonged lag-phase growth from 5 to 14 days (Fig. 5a). A similar phenotype was observed in the Δ*lamH* strain of *M. smegmatis* mc²155 (Supplementary Fig. 4d). To validate the role of *lamH* in this process, we also examined the growth kinetics of the complemented and

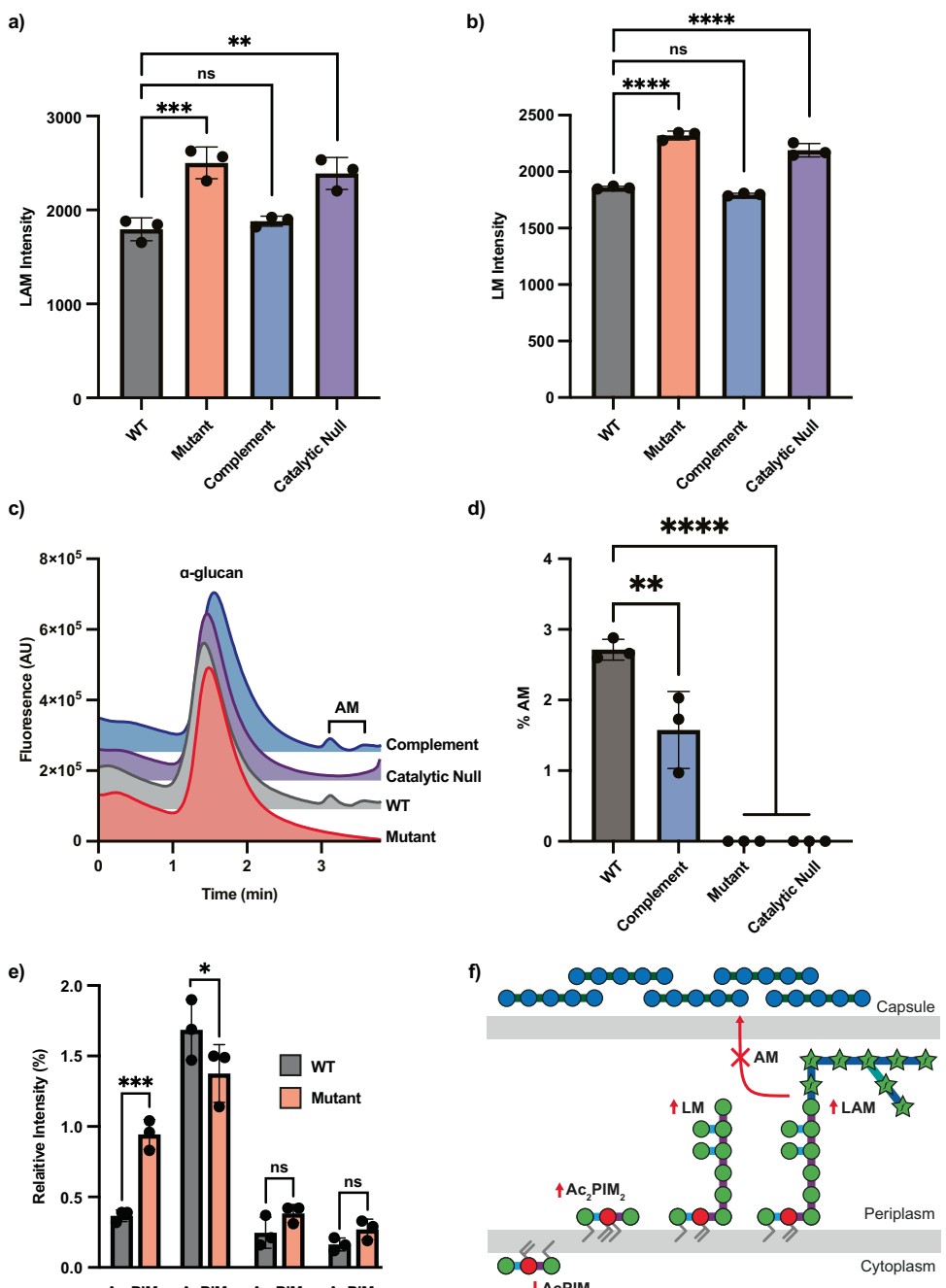

**Fig. 3 | A Δ*lamH* strain retains LAM in the cytoplasmic membrane and does not produce capsular AM. a** LM/LAM was harvested from the indicated *M. bovis* BCG strains grown in 7H9 broth with 0.05% Tween-80 at $OD_{600nm} = 0.6$ and analysed by SDS-PAGE followed by Pro-Q Emerald staining (Supplementary Fig. 1). Total fluorescence of LAM (**a**) and LM (**b**) for three biological replicates is presented with error bars indicating standard deviation and the centre of measure defined as the mean. **a** **$P = 0.0018$; ***$P = 0.0006$, **b** ****$P < 0.0001$. **c** The capsular material from the indicated strains was normalised based on wet mass, labelled with 2-AB and separated by size-exclusion chromatography with fluorescence detection (Ex = 320 nm, Em = 420 nm). Peaks were identified by analysing samples pre-digested with appropriate enzymes. **d** The α-glucan and AM peaks from (**c**) from three biological replicates were integrated, and the relative amount of AM was computed by dividing the total AM peak area by the sum of the α-glucan and AM peak areas. Error

bars indicate standard deviation. **$P = 0.003$; ****$P < 0.0001$. **e** The indicated *M. bovis* BCG strains were grown in 7H9 in the presence of $(1-{}^{14}C)$ acetic acid sodium salt until $OD_{600} = 0.6$. Polar lipid extracts were analysed from three biological replicates by two-dimensional TLC and annotated as per ref. 80. The relative intensity of each PIM species is reported with error bars representing standard deviation. *$P = 0.0306$; ***$P = 0.001$. **f** Schematic summary of phenotypic changes due to loss of *lamH*. WT−*M. bovis* BCG Danish 1331, Mutant−*M. bovis* BCG Danish 1331 *lamH::Himar1*, Complement−*M. bovis* BCG Danish 1331 *lamH::Himar1*, L5::pMV306-*lamH*, Catalytic Null−*M. bovis* BCG Danish 1331 *lamH::Himar1*, L5::pMV306-*lamH*$_{D96A}$. Significance was determined with a one-way ANOVA with Tuckey's post hoc test. *$P < 0.05$, **$P < 0.01$, ***$P < 0.001$, ****$P < 0.0001$. Source data are provided as a Source Data file.

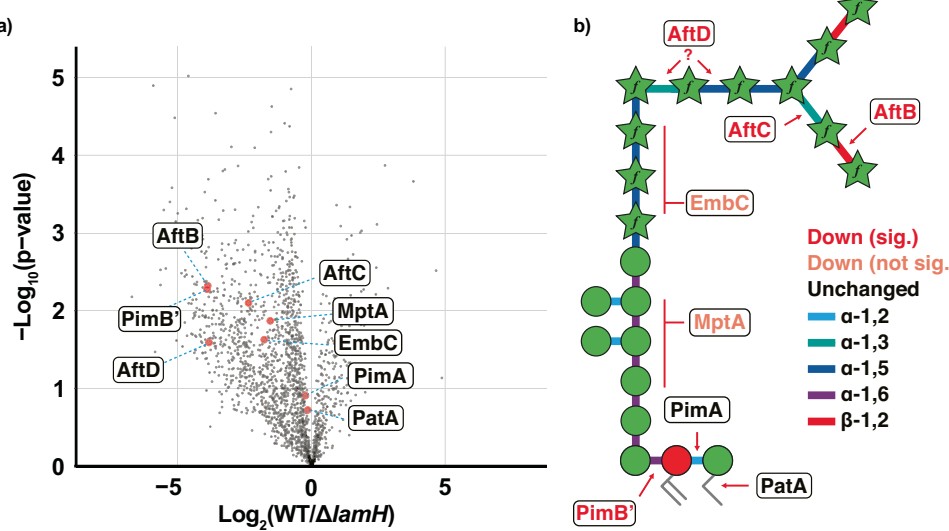

**Fig. 4 | Proteome analysis of *M. bovis* BCG Danish 1331 WT and Δ*lamH*. a** Volcano Plot of proteome alterations observed between *M. bovis* BCG and Δ*lamH* visualised as log₂(WT/Δ*lamH*) and -log₁₀(*P* values). Proteins of interest in the LAM biosynthetic pathway are denoted in red. Plots showing quantitation of statistically significant proteins are found in Supplementary Fig. 5. Significance was determined using two-tailed unpaired *t* tests, and multiple hypothesis correction was undertaken using a permutation-based FDR approach. The complete list of differentially expressed proteins is in Supplementary Data 1. **b** Eight proteins associated with LAM biogenesis were observed in the proteomics dataset. These have been annotated next to the bond on the simplified LAM structure for which they are understood to generate. Proteins coloured red meet statistical significance for being lower in abundance (Supplementary Fig. 5). Those in orange appear lower in abundance but do not meet statistical significance. Those in black have no change associated with their expression. Source data are provided as a Source Data file.

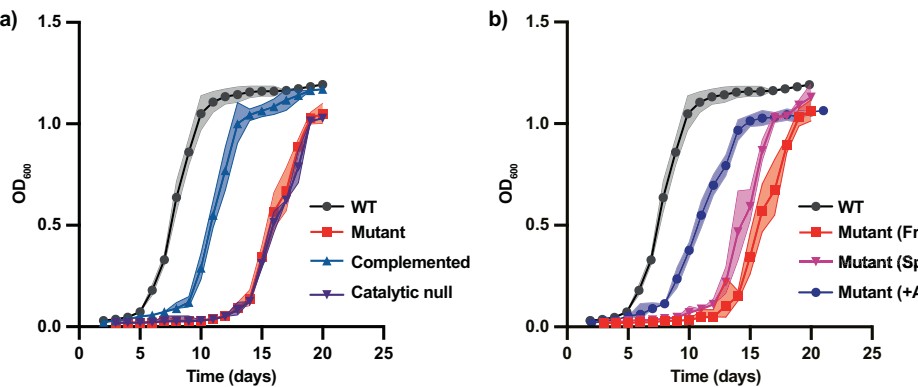

**Fig. 5 | LamH-catalysed AM turnover is required for efficient exit from lag phase. a** Growth kinetics of the indicated strains. Complemented—Δ*lamH* L5::pMV306-*lamH*, Catalytic null—Δ*lamH* L5::pMV306-*lamH_{D96A}*. **b** The Δ*lamH* strain was grown either in spent media from wild-type bacteria, fresh media or media supplemented with 0.5 mg/mL AM. The values for WT are included from (**a**) for comparison. WT— *M. bovis* BCG Danish 1331, Mutant—*M. bovis* BCG Danish 1331 *lamH::Himar1*, Complement—*M. bovis* BCG Danish 1331 *lamH::Himar1*, L5::pMV306-*lamH*, Catalytic Null—*M. bovis* BCG Danish 1331 *lamH::Himar1*, L5::pMV306-*lamH_{D96A}*. For all growth curves the shaded area represents the 95% confidence intervals of three biological replicates with the centre of measure defined as the mean and *n* = 3 biological replicates. Source data are provided as a Source Data file.

catalytically inactive strains described above. The results indicate that efficient exit from lag phase requires catalytically competent LamH in *M. bovis* BCG Danish 1331 (Fig. 5a). These findings indicate that under the conditions tested, *lamH* activity is necessary for efficient exit from lag-phase growth.

### LamH-derived arabinomannan triggers exit from lag phase

During growth in liquid media mycobacteria typically shed their capsule to the surrounding environment[56]. If capsular AM is a molecular signature for exit from the lag phase, we would expect that its accumulation in the media during liquid growth of wild-type bacteria might provide sufficient material in spent media to counter the growth defects of the Δ*lamH* mutant. Thus, we cultivated the mutant in spent media from mid-exponential wild-type culture. As shown in Fig. 4c, this

spent media partially rescued the lag-phase defect. Reasoning that the concentration of AM may be insufficient in spent media from mid-exponential bacteria, we enzymatically degraded LAM in vitro using recombinant LamH, and added it to fresh media at a 0.5 mg/mL. When cultivated in this supplemented media, the lag-phase defect of the mutant strain was rescued to a similar degree as for genetic complementation (Fig. 4c). These findings suggest that during the lag phase, LamH-driven production of AM may act as a molecular signal for the outgrowth of mycobacteria.

### Reduced expression of *lamH* impairs *M. tuberculosis* replication in macrophages

To assess the impact of *lamH* loss on *M. tuberculosis* host survival, we initially attempted to create a deletion mutant using specialised

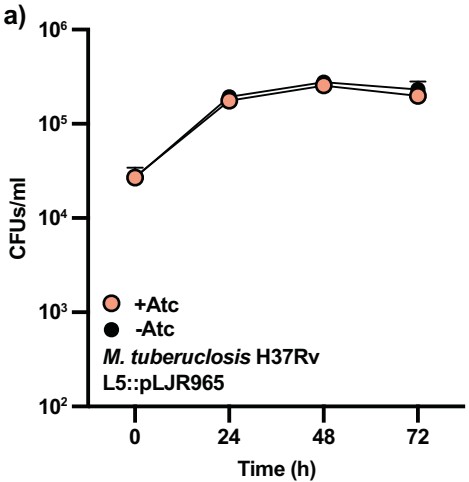 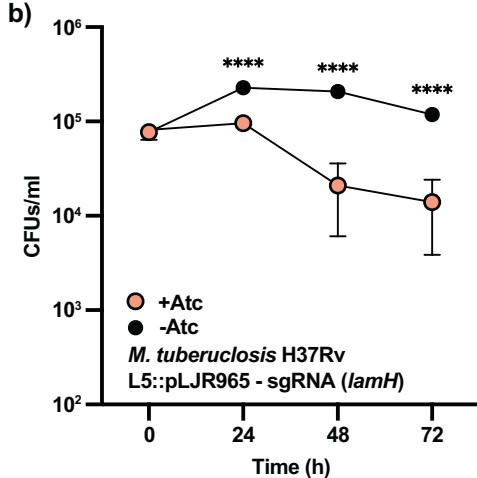

**Fig. 6 | Reduction of *lamH* expression leads to a decrease in fitness in macrophages. a** The parental *M. tuberculosis* H37Rv strain, including pLJR965 at the L5 site, was used to infect THP-1-derived macrophages in the presence and absence of ATc. Bacterial intracellular viability was examined at the indicated time points. **b** A strain carrying pLJR965 with a *lamH* guide RNA at the L5 site was also used to infect THP-1-derived macrophages similarly. This strain manifested a growth defect in the presence of ATc that was not observed when ATc was omitted from the culture medium. Data presented are mean and standard error and are representative of two independent experiments each with three biological replicates with similar results. Significant differences were calculated by a multiple *t* test using a two-stage linear step-up procedure of Benjamini, Krieger and Yekutieli, with Q = 1%. Each row was analysed individually without assuming a consistent standard deviation. Number of tests performed: 4. ****, 24 h *P* = 0.00079; 48 h *P* = 0.00007; 72 h *P* = 0.00024. Source data are provided as a Source Data file.

transduction. However, despite repeated attempts, we were unable to generate this mutant. This difficulty may stem from an extended lag-phase phenotype as observed in *M. bovis* BCG, or polar effects on nearby genes. To overcome this issue, we used CRISPRi to knock down *lamH*'s expression by incorporating a *lamH* guide RNA in the pLJR965 plasmid integrated at the L5 site in *M. tuberculosis* H37Rv[57]. Upon addition of anhydrotetracycline (ATc) to the resulting strain, the expression of *lamH* was reduced by approximately 95% compared to the same strain grown in the absence of ATc (Supplementary Fig. 6). In addition, we generated a parental strain transformed with an empty plasmid, which upon addition of ATc maintained the expression levels of *lamH* (Supplementary Fig. 6).

Subsequently, we assessed the requirement of *lamH* for intracellular survival within macrophages by infecting THP-1-derived macrophages with the *lamH* conditional mutant and the parental strain, in the presence or absence of ATc. We determined viable counts 1, 2 and 3 days post infection. Both the *lamH* conditional mutant and the parental strain retained their infective capacity without ATc (Fig. 6a). However, only the *lamH* conditional strain showed reduced viability in the presence of ATc, with significant reductions at all time points post infection (0.4-log reduction at 24 h; 1-log reduction at 48 h; 1-log reduction at 72 h) (Fig. 6b). These results indicate that *lamH* is important for mycobacterial fitness during infection.

## Discussion

Yokoyama and Ballou first reported the activity of an α-1,6-mannanase in mycobacteria in 1989, yet more than three decades later, the enzyme has eluded characterisation[58]. α-1,6-mannanases occur within the CAZy GH76 family, and according to the CAZy database, only 11 GH76 enzymes have been characterised, with none from pathogenic bacteria[59]. Our biochemical data identify LamH as an LM/LAM hydrolase that specifically targets the first α-1,6-linked mannose attached to the AcPIM$_2$ anchor of LM/LAM (Fig. 2h). We explored this specificity using structurally homogenous mannan from mutant strains. Our findings reveal that LamH does not digest mannan with α-1,2 backbone decorations, which are present on much of the mannan backbone of LAM distal from the AcPIM$_2$ anchor[7]. This specificity is likely influenced by the predicted β-hairpin capping the active site

(Fig. 1b). Comparison with the Aman6 structure suggests that α-1,6-mannotetraose should fit within the LamH active site. However, the capping β-hairpin and active site could potentially clash with substituents on mannose residues in the -1 and -2 positions. Further investigations will be necessary to resolve the precise substrate recognition mechanism of LamH.

Our biochemical results indicate that AcPIM$_2$ is the predominant, if not exclusive, anchor of LM and LAM in *M. bovis* BCG. This is consistent with previous studies, although we did not observe variation in the acylation status of these molecules[60]. The generation of AcPIM$_2$ upon digestion with LamH suggests a possible pathway for recycling this lipid anchor into new rounds of LM/LAM and AM biogenesis. This is reminiscent of known mechanisms by which mycobacteria recycle peptidoglycan components and trehalose mycolates[61–63]. Given its high specificity, LamH will likely be a valuable tool for exploring the structure of the PIM anchor of LM/LAMs isolated from other mycobacterial species and, importantly, clinical isolates. Recently, the Jackson laboratory identified an Aman6 isozyme from *Bacillus circulans* for a similar purpose, which demonstrated activity against LAM[64]. However, unlike the high specificity of LamH, this enzyme appears to hydrolyse LM/LAM at multiple positions within the mannan chain, posing challenges in the analysis of reaction products.

Three non-exclusive hypotheses have been proposed to explain how LM, LAM or their by-products are released to the host. First, LM/LAM might be extracted from the cytoplasmic membrane and trafficked to the outside of the cell through an active secretion system. The lipoprotein LprG has been implicated as a potential carrier protein for LM/LAM, and loss of *lprG* leads to reduced surface exposure of LM/LAM[65,66]. However, conflicting reports suggest that the physiological role of this protein is to transport triacylglycerols, though these two roles for LprG may not be mutually exclusive[67,68]. Alternative mechanisms for active LM/LAM secretion remain to be discovered. The second hypothesis is the production of vesicles by the bacteria, which are reported to contain LM/LAM[69,70]. A third hypothesis for LM/LAM/AM secretion is that the host primarily recognises capsular AM, rather than LAM itself. For example, the production of capsular AM has been shown to drive the production of protective antibodies[22,71–73]. It is likely that these three hypotheses

are not mutually exclusive, and that multiple mechanisms contribute to LM/LAM/AM exposure in the host.

To our knowledge, this work provides the first mechanistic insight into the generation of capsular AM (Fig. 3c). Our data also provide evidence that the generation of this material is important for *M. tuberculosis* fitness in macrophages. While we could not delete *lamH* in *M. tuberculosis* H37Rv, CRISPRi-mediated silencing substantially reduced its expression resulting in approximately tenfold growth reduction in macrophages, aligning with prior findings indicating a growth advantage in macrophages conferred by expression of *M. tuberculosis lamH* in *M. smegmatis* mc[2]155[74]. This suggests that LamH-driven AM production could be a promising anti-virulence target.

While LM and LAM play important roles during pathogenesis, most mycobacteria, despite generating these glycolipids, are considered non-pathogenic. Recent studies from the Morita group identified a role for LAM in septation, akin to lipoteichoic acids in other bacteria[55,75]. Our analysis of ΔlamH mutants in *M. bovis* BCG and *M. smegmatis* underscores the critical role of LamH in maintaining LM/LAM homoeostasis and driving capsular AM expression. This suggests bacterial monitoring of LM/LAM or AM levels in the cell and regulation of biosynthetic enzymes through an as-yet-unknown signalling pathway. The profound lag-phase defect observed upon loss of *lamH* in *M. bovis* BCG supports this hypothesis, with defect repair requiring the LamH catalytic activity or AM supplementation, indicating AM's role in promoting outgrowth under these conditions. Use of AM as a carbon nutrient can be excluded as *M. tuberculosis* H37Rv has previously been shown to be unable to utilise D-arabinose or D-mannose as carbon sources[76]. While our data supports a role for AM in a signalling process, it is possible that the signalling molecule is a fragment derived from additional processing of AM, perhaps mediated by the recently discovered mycobacterial endo-D-arabinanases[21]. This could imply that the abundance of AM or degradation fragments could report on cell growth and division, offering insights into bacterial metabolic status. This model could explain mycobacterial growth phase transitions, explaining the population-level shift to exponential growth in response to AM abundance.

## Methods

### Bioinformatic analysis
Genomes for diverse members of the Mycobacteriales were downloaded from NCBI and used to generate a custom BLAST database in Geneious Prime 2023.1.1. A list of the genome accessions used is found in Supplementary Data 2. Rv0365c was used as a query in a BLAST search of this database, yielding high-confidence homologs from all species[38]. This list of proteins was filtered to include a single highest-quality match per species and submitted to NGPhylogeny.fr using the PHYML/OneClick tool; the tree was then manually coloured to identify individual genera[37]. The Alphafold 2.0 structural prediction of Rv0365c was accessed at Uniprot (O06315_MYCTU) and aligned to PDB 5AGD using LSQKab in Coot 0.9.9.6[28,77]. Figures were prepared using ChimeraX 1.6[78].

### Strains and growth conditions
Unless stated otherwise, all chemicals and reagents were purchased from Sigma-Aldrich. Strains used were *Escherichia coli* T7 Shuffle (New England Biolabs) to express Rv0365c, Rv0365c D96A and BT3792; *S. cerevisiae* Mnn1, Mnn2, and Mnn5[25]; *M. bovis* BCG Danish 1331 WT and *M. bovis* BCG Danish 1331 (*BCGDAN_0378::Himar1*)[40]; *M. smegmatis* mc[2]155 WT[79] and *M. smegmatis* mc[2]155 ΔMSMEG_0740. *E. coli* strains, up to 100 mL, were grown in lysogeny broth. *E. coli* cultures used for protein purification were grown in terrific broth at 37 °C with agitation. With agitation, *S. cerevisiae* cultures were grown in yeast extract-peptone-dextrose media at 30 °C. *M. bovis* BCG Danish 1331 was grown in 7H9 media for liquid cultivation or on 7H10-OADC agar for solid growth at 37 °C with 5% $CO_2$. *M. smegmatis* mc[2]155 was grown in

Middlebrook 7H9-ADC media at 37 °C. *M. tuberculosis lamH* conditional mutant was generated using CRISPRi technology as previously described[57]. Briefly, two oligonucleotides (5´-TGGCTAACAGCTA TTACGACTCCC-3´) complementary to Rv0365 were synthesised, annealed, and cloned into pLJR965 plasmid. The selection of the sgRNAs was based on a theoretical degree of repression of 100%, according to the PAM sequences. The vectors were transformed into *E. coli* and extracted to confirm the presence of the sgRNA by Sanger sequencing using the primer, 5'-TTCCTGTGAAGAGCCATTGATAATG-3'. The resulting construct was electroporated into the parental mycobacterial strain H37Rv and selected on 25 µg/ml kanamycin. The parental strain carrying the empty vector (without a targeting sgRNA) was used as a negative control.

To monitor the growth of *lamH* conditional mutant and the parental strain harbouring empty pLJR965 plasmid, mid-log liquid cultures were diluted to an optical density at 600 nm ($OD_{600}$) of 0.05 in 7H9-ADC + 0.05% tyloxapol with or without 200 ng/mL ATc and incubated without shaking at 37 °C. $OD_{600}$ was monitored for 15 consecutive days.

### ORBIT-mediated mutagenesis of *M. smegmatis* mc[2]155
ORBIT-mediated mutagenesis of MSMEG_0740 was carried out according to published protocols[43]. Twenty mL of *M. smegmatis* mc[2]155 [pKM461] cells were grown in 7h9 media containing 25 µg/ml kanamycin at 37 °C in a shaking incubator. At an $OD_{600}$ of 0.5, ATc was added to the culture at a final concentration of 500 ng/mL. This was incubated for a further 3 h at 37 °C. The culture was washed twice in 20 mL of sterile 10% cold glycerol. Following the second wash, cells were collected via centrifugation and resuspended in 2 mL of 10% cold glycerol. On ice, 380 µL of the electrocompetent cells, 200 ng of the pKM464 payload plasmid and 1 µg of the targeting oligonucleotide (gctactcctcatcctcgttctcgtcgtgtcccacttcgccgtctccggcgccgtttggtctccgg cagcggtttgtctggtcaaccaccgcggtctcagtggtgtacggtacaaacctgagttcggtg atcgcagcttcggcactggccgcccggttggcccatagctgatccatggcaacgatcctgcc) were added to a 2 mm gap width electroporation cuvette. The cells were then electroporated at 2.5 kV before overnight incubation (37 °C) in 7h9 media with 0.05% Tween-80. The next day, 0.5 mL of culture was spread onto 7H10 plates containing hygromycin (50 µg/mL) and incubated for 3–4 days at 37 °C to select transformants. Selected colonies were analysed by PCR.

### Quantification of mRNA of lamH by qPCR
Cultures were grown to log phase and then diluted back to an $OD_{600}$ of 0.02 in the presence or absence of 200 ng/ml ATc. Target knockdown was allowed to proceed for four days. Next, cells were harvested by centrifugation, resuspended in TRIzol (Thermo Fisher), and disrupted by bead-beating (Lysing Matrix B, MP Biomedicals). Total RNA was isolated by RNA miniprep (Zymo Research). cDNA was prepared with random hexamers per manufacturer instructions (Life Technologies Superscript IV). cDNA levels were quantified by quantitative real-time PCR (qRT-PCR) on an Applied Biosystems light cycler (Applied Biosystems) using a SYBR Green PCR Master Mix (Thermo Fisher Scientific) using specific primers. Signals were normalised to the housekeeping *sigA* transcript and quantified by the ΔΔCt method.

### Radiolabelling of mycobacterial lipids
*M. bovis* BCG Danish 1331 cultures were grown until an $OD_{600}$ of 0.2 was reached. Cultures were radiolabelled with the addition of 10 µCi/ml acetic acid sodium salt [1-$^{14}$C] (specific activity 50–62 mCi /mmol; 1850–2294 MBq/mmol; Perkin Elmer) and further incubated until an $OD_{600}$ of 0.8 was reached.

### Protein expression and purification
The expression plasmid for Rv0365c was synthesised and codon-optimised by Twist Biosciences in a pET28a vector, including residues

2–376 with an N-terminal thrombin cleavable histidine tag. The BT3792 expression plasmid was previously reported[25]. Recombinant proteins were expressed in competent *E. coli* T7 Shuffle cells. Site-directed mutants were generated using the New England Biolabs Q5 mutagenesis kit. For purification of Rv0365c, cultures were grown until an $OD_{600}$ of 0.6 was reached; at this point, 0.1 mM IPTG was added to induce protein expression. Cells were further incubated for 16 h at 14 °C. Cells were harvested by centrifugation at 6000×*g* for 20 min at 4 °C and resuspended in 100 mM HEPES pH 7.5, 300 mM NaCl, and 5 mM imidazole pH 7.5. The resuspended pellet was stored at −20 °C until further use. Pellets were thawed, and 5% glycerol and 1% Tween20 were added. Next, 1 mg/mL deoxyribonuclease from bovine pancreas (Sigma-Aldrich) was added to the resuspension and incubated on ice for 30 min. The cell resuspension was lysed with three passages through a French pressure cell. Cell debris was pelleted by centrifugation at 40,000×*g* for 45 min at 4 °C. Enzymes were purified using immobilised metal affinity chromatography (IMAC) on nickel Sepharose resin in a gravity column. Bound protein was eluted from the column with increasing concentrations, 5 mM to 500 mM of imidazole washes. Positive fractions were determined by SDS-PAGE and dialysed into 100 mM HEPES pH 7.5, 300 mM NaCl buffer at 4 °C. The protein was concentrated to a final volume of 500 μL using a 30 kDa molecular weight cut-off protein concentrator (Thermo Scientific).

For purification of BT3792, cultures were grown until an $OD_{600}$ of 0.6 was reached. Protein expression was induced by adding 0.2 mM IPTG, and cultures were incubated for a further 16 h at 16 °C with 180 rpm shaking. As before, cell pellets were harvested by centrifugation at 6000×*g* for 20 min at 4 °C and resuspended in 150 mM Tris pH 8.0, 300 mM NaCl, and 20 mM imidazole pH 8.0. Cells were stored at −20 °C until use. The pellets were thawed, and 1 mg/mL deoxyribonuclease from bovine pancreas (Sigma-Aldrich) was added to the resuspension and incubated on ice for 30 min. Cells were lysed with three passages through a French pressure cell. Cell debris was pelleted by centrifugation at 40,000×*g* for 45 min at 4 °C. Enzymes were purified by IMAC using nickel Sepharose resin in a gravity column. Bound enzymes were eluted from the column with an imidazole gradient ranging from 20 mM to 500 mM. Positive fractions were identified by SDS-PAGE and dialysed for 16 h at 16 °C into 150 mM Tris pH 8.0, 300 mM NaCl buffer. As before, the protein was concentrated to a final volume of 500 μL using a 30 kDa molecular weight cut-off protein concentrator (Thermo Scientific).

### Enzyme assays
Purified protein, at a final concentration of 1 μM, was incubated with 1 mg/ml substrate at 37 °C for 16 h (unless stated otherwise) in 100 mM HEPES buffer pH 7.5. Reactions were heat-inactivated by incubation at 100 °C for 10 min and analysed by thin-layer chromatography.

### Thin-layer chromatography (TLC)
Samples were spotted onto a TLC plate (Merck, TLC Silica Gel 60 $F_{254}$) and separated until the solvent front reached 5 mm from the top of the plate. The TLC plate was dried and either stained and heated or exposed to X-ray film for radioactive samples.

### TLC solvent systems
For mannans, samples were separated in n-butanol:acetic acid:water (2:1:1 v/v/v), sprayed with orcinol (5 g orcinol in 375 mL methanol, 107 mL water, 16.2 mL concentrated sulfuric acid) and charred to reveal products. To analyse LAM, samples were separated in chloroform: methanol: water (65:25:3 v/v/v), stained with either orcinol or molybdophosphoric acid (MPA) (10 g phosphomolybdic acid in 100 mL absolute ethanol), and charred by heating. Two-dimensional TLC was used for the analysis of PIMs. The samples were separated in the first direction in chloroform:methanol:water (60:30:6 v/v/v) and in the second direction in chloroform:acetic acid:methanol:water (65:25:3:6 v/v/v/v). TLCs were annotated as per ref. 80.

### Purification of α-mannan from *S. cerevisiae*
Purification of α-mannan from *S. cerevisiae* was performed as previously described[25]. Eight litres of the desired *S. cerevisiae* strain were grown at 37 °C for 24 h. Pellets were harvested by centrifugation at 6000×*g* for 20 min. The pellets were then pooled and stored at −20 °C. The pooled cell pellets were resuspended in 20 mL 0.02 M citrate buffer pH 7.0 and then autoclaved for 90 min at 121 °C. The sample was centrifuged at 6000×*g* for 10 min, and the supernatant was collected. The remaining pellet was resuspended in 75 mL of 0.02 M citrate buffer pH 7.0 and autoclaved at 121 °C for 90 min. The sample was centrifuged at 6000×*g* for 10 min, and the resulting supernatant was pooled with that previously collected. A volume of 2× Fehling's reagent, equal to that of the supernatant, was measured and heated to 40 °C. The supernatant was carefully added to the 2× Fehling's reagent and stirred vigorously at 40 °C for 1 h. The mixture was centrifuged at 6000×*g* for 10 min to harvest the pellet. The pellet was dissolved in 8 mL of 3 M hydrochloric acid before 100 mL of methanol: acetic acid solution (8:1 v/v) was added and stirred for 1 h at room temperature. The mixture was centrifuged at 12,000×*g* for 15 min, and the pellet was collected. The pellet was resuspended in 20 mL methanol: acetic acid solution (8:1 v/v) and centrifuged again. This process was repeated until the pellet was colourless. The pellet was then left to dry at room temperature overnight. The dry pellet was resuspended in 20 mL dH₂O and then dialysed against 4 L of dH₂O for 24 h. The mixture was lyophilised to complete dryness.

### Purification of mycobacterial glycolipids
One litre of mycobacterial culture was grown until an $OD_{600}$ of 0.8 was reached, and cells were harvested by centrifugation at 6000×*g* for 10 min. The pellet was resuspended in 20 mL PBS, 0.1% Tween-80, and lysed by bead-beating. The cell lysate was transferred to a Teflon-capped glass tube, and an equal volume of phenol was added. The mixture was heated to 85 °C and incubated for 2 h with regular mixing. The aqueous phase was separated from the phenol phase by centrifugation at 4000×*g* for 10 min and transferred to a fresh tube. The phenol wash is repeated twice more. The glycan mixture was dialysed exhaustively against tap water overnight. Following this, the mixture was further dialysed against ddH₂O for 2 h and then lyophilised. The extracted glycolipids were resuspended in ddH₂O normalised to 1 mg wet cell mass/mL ddH₂O. These samples were analysed via SDS-PAGE with Pro-Q glycolipid staining. Twenty uL of glycolipid sample and 5 μL of loading dye were mixed, and the glycolipids were separated on a 4–20% precast gel and then incubated in 100 ml 50% methanol, 5% acetic acid for 30 min to fix the gel. This was repeated once more. The gel was washed three times in 3% acetic acid in water for 15 min. Following this, the gel was oxidised in a periodic acid solution for 30 min. As before, the gel was washed with 3% acetic acid. Following this, the gel was stained with the Pro-Q staining solution, washed once more, and imaged using a BioRad Gel Doc at 300 nm and the files were analysed in ImageLab 6.1.

### Analysis of mannotetraose by ion chromatography-mass spectrometry (IC-MS)
Mannotetraose was isolated from an overnight digest of 1 g of mannan derived from *S. cerevisiae* by BT3792, as described in ref. 25. Oligosaccharide fractions were separated on two BioGel P2 columns in series, run in distilled water at 0.2 mL min⁻¹. Fractions were analysed via TLC and then pooled and freeze-dried. 50 μM of the likely mannotetraose was analysed on a ThermoFisher IC-MS system comprised of an ICS-6000 liquid chromatography system with an ERD 500 suppressor and pulsed amperometric detection, coupled to an Orbitrap

Exploris 240 mass spectrometer. IC elution on a CarboPac Pa300 column was run at 0.25 mL min⁻¹ as follows: 0-10 min isocratic; 100 mM NaOH. 10–50 min; gradient of 0–200 mM sodium acetate in 100 mM NaOH; 50–60 min, 300 mM sodium acetate wash; 60–70 min, 300 mM NaOH wash; 70–85 min re-equilibration in 100 mM NaOH.

Mass spectrometry was performed in negative mode using H-ESI with the following parameters: negative ion voltage, 3400 V; ion transfer tube temperature, 325 °C; vaporiser temperature, 350 °C; and scan range ($m/z$), 300–2500. The orbitrap was calibrated with FlexMix calibration solution. Data analysis was performed using ThermoFisher Freestyle 1.8 SP2.

### Polar lipid extraction

Mycobacterial cultures were grown until an $OD_{600}$ of 0.8 was reached and harvested by centrifugation at 6000×g for 10 min. The pellets were resuspended in PBS, 0.1% Tween-80, and transferred to a Teflon-capped glass tube. 2 mL methanol: 0.3 % NaCl (100:10 v/v) and 2 mL petroleum ether (60–80) was added to the pellet and mixed for 24 h on a rotator. The sample was centrifuged at 1900×g for 10 min to form a bilayer, and the upper layer was transferred to a fresh tube. An additional 2 mL of petroleum ether is added to the remaining lower layer and mixed for 1 h. The sample was centrifuged as before, and the upper layers pooled. The petroleum ether extracts were dried under a stream of nitrogen to give the non-polar lipids. To the remaining lower layer, 750 μL of chloroform: methanol: 0.3% NaCl (9:10:3 v/v/v) was added and mixed for 2 h. The sample was centrifuged at 3000×g for 10 min, and the supernatant was transferred to a fresh tube. The pellet was resuspended in 950 μL chloroform:methanol:0.3% NaCl (5:10:4 v/v/v) and mixed for 30 min. The sample was centrifuged at 1500×g for 5 min, and the supernatant was pooled with that from the previous step. The polar lipids were extracted by adding 1 mL chloroform and 1 mL 0.3 % NaCl to form a bilayer. The lower, polar lipid-containing layer was transferred to a fresh tube and dried under a stream of nitrogen.

### Capsular polysaccharide extraction and analysis

*M. bovis* BCG Danish cultures were grown on 7H10 agar plates for four weeks. Subsequently, cells were scraped from the plates, and a cell mass was measured. The cells were then resuspended in 10 mL of dH₂O and vortexed at the lowest speed setting on a Vortex-Genie 2 for 1 min to shed capsular material. Cells were harvested by centrifugation at 2000×g for 10 min. The capsule-containing supernatant was collected and filtered using a 0.45 μM Millipore filter to ensure no bacterial debris remained. The capsular extract was frozen and lyophilised to complete dryness.

Fluorescent labelling of the capsular polysaccharide extracts was carried out following the methods of Ruhaak et al.[81]. In brief, 50 μl of capsular material was mixed with 25 μl of freshly prepared label (48 mg/mL 2-aminobenzamide (Ludger) in DMSO/acetic acid (85:15 v/v)). Next, 25 μl of 1 M 1-picolane-borane in DMSO was added to achieve a final volume of 100 μl. Subsequently, the mixture was incubated at 65 °C for 2 h. The samples were then allowed to cool to room temperature before being analysed.

The fluorescently labelled capsular extracts were analysed using a Phenomenex BioZen 1.8 μM size-exclusion chromatography-2 column (300 × 4.6 mm, 00H-4769-E0) at room temperature on a Dionex Ultimate 3000 uHPLC controlled with Chromeleon Chromatography Data System Software 7.2 SR4. The mobile phase was 0.1 M phosphate buffer pH 6.8 with 0.025 % sodium azide, flow rate = 0.400 mL/min. Glycans detected with fluorescence detection (Ex = 320 nm Em = 420 nm), sample volume injected = 2.5 μL. Peak identities were confirmed by digestion with α-amylase (Sigma-Aldrich), BT3792, or GH183$_{Mab}$ under the same conditions used for LamH assays[21,25].

### Growth kinetics determination

Bacterial growth was recorded by taking $OD_{600}$ readings daily until the stationary phase was reached. Cultures were grown static, in triplicate, in 7H9 media at 37 °C with 5% CO₂ using Corning culture flasks (Sigma-Aldrich, CLS431082). Starter cultures were grown until mid-log phase ($OD_{600}$ = 0.6) was reached and subsequently diluted to an $OD_{600}$ of 0.01 in 10 mL 7H9 media, and growth was recorded every 24 h (*M. bovis* BCG) or 1 h (*M. smegmatis*).

### Sample preparation for proteomic analysis

*M. bovis* BCG cultures obtained at the mid-log phase were washed once in PBS to remove media proteins before being boiled in 1 mL 100 mM Tris pH 8.5 with shaking. The samples were centrifuged at 17,000×g for 10 min to pellet cell debris. 180 μL of protein sample was transferred to an acetone-resistant tube, and 20 μL 1 M NaCl and 800 μL ice-cold acetone were added. The samples were then incubated overnight at −20 °C. The samples were centrifuged at 1100×g for 10 min at 0 °C, and the acetone was removed. The pellet was resuspended in 200 μL of Milli-Q, and an additional 800 μL ice-cold acetone was added. This was incubated for a further 4 h at −20 °C. The samples were centrifuged for a final time at 2400×g for 20 min at 0 °C, the acetone was removed, and the pellet was allowed to dry.

Acetone-precipitated proteome samples were solubilised in 4% SDS, 100 mM Tris pH 8.5 by boiling for 10 min at 95 °C. The protein concentrations were assessed using a bicinchoninic acid protein assay (Thermo Fisher Scientific), and 100 μg of each biological replicate was prepared for digestion using Micro S-traps (Protifi, USA) according to the manufacturer's instructions. Samples were reduced with 10 mM DTT for 10 min at 95 °C and then alkylated with 40 mM IAA in the dark for 1 h. Samples were acidified to 1.2% phosphoric acid, diluted with seven volumes of S-trap wash buffer (90% methanol, 100 mM tetra-ethylammonium bromide pH 7.1), then loaded onto S-traps and washed three times with S-trap wash buffer. Samples were digested overnight with Trypsin (1:100 protease:protein ratio, Solu-Trypsin, Sigma) within 100 mM Tetraethylammonium bromide pH 8.5. Following overnight digestion, peptides were collected by centrifugation using washes of 100 mM Tetraethylammonium bromide, 0.2% formic acid, and 0.2% formic acid / 50% acetonitrile. Samples were dried down and further cleaned up using C18 Stage tips to ensure the removal of any particulate matter[82,83].

### Reverse-phase liquid chromatography-mass spectrometry

Proteome samples were resuspended in Buffer A* (0.1% trifluoroacetic acid, 2% acetonitrile) and separated using an Ultimate 3000 UPLC (Thermo Fisher Scientific) equipped with a two-column chromatography set-up composed of a PepMap100 C18 20 mm × 75 μm trap and PepMap C18 500 mm × 75 μm analytical column (Thermo Fisher Scientific). Samples were concentrated onto the trap column at 5 μL/min for 6 min with Buffer A (0.1% formic acid, 2% DMSO) and then infused into an Orbitrap Fusion™ Eclipse™ Tribrid™ mass spectrometer equipped with a FAIMS Pro interface (Thermo Fisher Scientific) at 300 nl/min via an analytical column. 89-min analytical runs were undertaken by altering the buffer composition from 2% Buffer B (0.1% formic acid, 77.9% acetonitrile, 2% DMSO) to 28% B over 70 min, then from 28% B to 40% B over 9 min, then from 40% B to 80% B over 3 min. The composition was held at 80% B for 2 min, then dropped to 2% B over 0.1 min before being held at 2% B for another 2.9 min. Each biological replicate was analysed using four different FAIMS compensation voltages (CVs, -25, -35, -45 and -65) in a data-dependent manner switching between the acquisition of a single Orbitrap MS scan (450–2000 $m/z$, maximal injection time of 50 ms, an AGC set to a maximum of $1 \times 10^6$ ions and a resolution of 60k) every 3 s followed by Orbitrap MS/MS HCD scans (using the "Auto" mass range setting, a NCE of 25;32;40%, maximal injection time of 120 ms, an AGC set to a

maximum of 500% and a resolution of 30k) as well as a Orbitrap EThcD scan (NCE 25%, maximal injection time of 120 ms with an AGC of 500% and a resolution of 30,000) undertaken on each precursor.

FragPipe version 19[84–88] was used to process the resulting proteome dataset with proteins identified by searching against the *M. bovis* BCG Danish 1331 proteome (NCBI accession: CP039850.1), allowing carbamidomethyl (57.0214 Da) of cystines as a fixed modification as well as oxidation of methionine (15.9949 Da), N-terminal acetylation (42.0106 Da) in addition to hexose (162.0528 Da) and 2*Hexose (324.1056 Da) on serine and threonine residues as variable modifications. The resulting datasets were filtered using the default FragPipe parameters of 1% peptide/protein level false discovery rates. IonQuant was utilised for quantitative proteome comparisons, and the default parameters enabled matching between runs across biological replicates. The resulting combined MSfragger protein level output was processed using Perseus (version 1.6.0.7). The missing values were imputed based on the total observed protein intensities with a range of 0.3 σ and a downshift of 1.8 σ[89]. Tables of imputed and non-imputed values are found in Supplementary Data 1a and 1b, respectively. Statistical analysis was undertaken in Perseus using two-tailed unpaired *t* tests.

### Gene ontology analysis

Matching of protein homologs between *Mycobacterium tuberculosis* variant bovis strain Danish 1331 (NCBI: PRJNA494982) and *Mycobacterium tuberculosis* strain H37Rv (Uniprot proteome: UP000001584) was undertaken using the proteome comparison tool of PATRIC, the bacterial bioinformatics database and analysis resource[90]. Gene Ontology (GO) terms associated with (*Mycobacterium tuberculosis* strain ATCC 25618/H37Rv proteome: UP000001584) were used to allow Enrichment analysis of proteome changes using Fisher exact tests within Perseus[89] using a 5% FDR and a Benjamini–Hochberg multiple hypothesis correction.

### Macrophage infection

THP-1 cells were seeded in 96-well plates for macrophage infection and differentiated using 50 nM phorbol myristate acetate (PMA). After 24 h of differentiation, PMA was removed, and cells were left to rest for 24 h. Cells were infected with *M. tuberculosis* H37Rv L5::pLJR965 or the same strain with a *lamH* sgRNA at 1:5 MOI (macrophage: bacteria) in RPMI medium containing 10% foetal bovine serum (FBS). After 4 h of incubation, cells were washed with pre-warm 1× PBS and replenished with RPMI-heat-inactivated(HI) FBS containing gentamicin (50 μg/ml) for 1 h to kill extracellular bacteria. Subsequently, cells were washed and maintained in RPMI-HI-FBS medium containing 200 ng/ml ATc throughout the experiment. At different time points, cells from three wells were harvested using 1× PBS + 0.1% Triton X100 and incubated on ice for 5 min to release intracellular bacteria. Cell suspensions were serially diluted in 1× PBS and spread on 7H10 agar plates for CFU enumeration after 4 weeks of incubation at 37 °C.

### Statistics and reproducibility

No statistical method was used to pre-determine sample sizes, and no data were excluded from analyses. The experiments were not randomised, and the investigators were not blinded to allocation during experiments and outcome assessment. The nature of the experiments precluded this. All plots were prepared in Graphpad Prism 10. The verification of reproducibility was achieved through the use of three or more biological replicates in all experiments where possible.

### Reporting summary

Further information on research design is available in the Nature Portfolio Reporting Summary linked to this article.

## Data availability

The mass spectrometry proteomics data has been deposited in the Proteome Xchange Consortium via the PRIDE partner repository with the data set identifier PXD042653[91]. All data for the manuscript is provided either in the Supplementary Materials or in the Source Data file. Source data are provided with this paper.

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

## Acknowledgements

The Biotechnology and Biosciences Research Council supported this work (grant BB/S010122/1 and BB/X00841X/1 to PJM and BB/X016749/1 to ECL). IC-MS equipment at Newcastle was funded by BB/W01954X/1). We also thank the Wellcome Trust for supporting this work (226644/Z/22/Z to PJM, S.W., and E.L., 209437/Z/17/Z to A.L.L., and a studentship to S.B.) G.S.B is supported by the Medical Research Council (MR/S000542/1). This work was supported in part by the MICINN grant PID2022-138694OB-I00 and the NIH grant R01GM148075 to M.E.G. We thank the Australian Research Council for a Future Fellowship to NES (FT200100270) and ARC Discovery Project Grants to NES and SJW (DP210100362, DP210100233, DP210100235). We thank the Melbourne Mass Spectrometry and Proteomics Facility of The Bio21 Molecular Science and Biotechnology Institute for access to MS instrumentation. We thank members of the Birmingham mycobacteriology group for helpful discussions.

## Author contributions

Aaron Franklin—methodology, validation, formal analysis, investigation, data curation, visualisation, writing—first draft, writing—review and editing. Vivian C. Salgueiro—methodology, validation, formal analysis, investigation, data curation and visualisation. Abigail Layton— methodology, validation and resources. Rudi Sullivan—methodology, validation and resources. Todd Mize—methodology, validation, formal analysis, investigation and data curation. Lucía Vázquez-Iniesta—

methodology, validation, formal analysis, investigation, data curation and visualisation. Samuel Benedict—methodology, resources and validation. Sudagar S. Gurcha—methodology, resources and validation. Itxaso Anso—methodology, validation, formal analysis and investigation. Gurdyal S. Besra—resources, supervision, funding acquisition and writing— editing. Manuel Banzhaf—resources, supervision, funding acquisition and writing—editing. Andrew Lovering—methodology, validation, formal analysis, writing—review and editing, supervision and funding acquisition. Spencer J. Williams—methodology, resources, validation, formal analysis, writing—review and editing, supervision and funding acquisition. Marcelo E. Guerin—methodology, resources, validation, formal analysis, writing—review and editing, supervision and funding acquisition, Nichollas E. Scott—methodology, resources, validation, formal analysis, writing—review and editing and funding acquisition. Rafael Prados-Rosales—conceptualisation, methodology, resources, validation, formal analysis, writing—review and editing, supervision and funding acquisition. Elisabeth C. Lowe—conceptualisation, methodology, validation, formal analysis, data curation, writing—review and editing, visualisation, supervision and funding acquisition. Patrick J. Moynihan—conceptualisation, methodology, validation, formal analysis, data curation, writing—first draft preparation, writing—review and editing, visualisation, supervision and funding acquisition.

## Competing interests

The authors declare no competing interests.
