## [Peer Review File · Nature Communications]

The mycobacterial glycoside hydrolase LamH enables capsular arabinomannan release and stimulates growthREVIEWER COMMENTS

Reviewer #1 (Remarks to the Author):

This study reports the identification of LamH, a mannosidase that releases arabinomannan from LAM. The presence of arabinomannan and mannan in the mycobacterial capsule has been known for a long time, but it remained unknown if there is an enzyme that actively engages in hydrolyzing LAM to release these capsular glycans. This study elegantly fills in this gap using bioinformatic, genetic, and biochemical approaches. It is a well-written manuscript, and the study is of high significance to the field of mycobacteriology and glycobiology.

Below, I have several comments to improve the clarity of the manuscript.

Major Comments:

The AM acting as a signalling molecule is a farfetched concept. AM may be acting simply as an absorbent to remove a molecule that inhibits the exit of cells from lag phase. Such an effect of AM would be considered an indirect effect of accumulating AM. I suggest the authors to provide more solid evidence to prove the role of AM as a signalling molecule, or tone down their claims.

The growth conditions are unclear. For some experiments, authors grew cells on 7H10 agar plate and extracted AM from cells by vortexing. However, for experiments in Fig. 5 and Extended Data Fig. 3, the cells were grown in 7H9 broth planktonically. Does 7H9 broth contain ADC and 0.05% tyloxapol (like the authors did to grow Mtb)? If *M. smegmatis* was grown in a complete 7H9 medium, why does the culture go into stationary phase at the OD600 of ~ 1.2 in Extended Data Fig. 3d? In 7H9, *M. smegmatis* can normally reach to OD600 = 4~5.

Assuming that the culture medium contained either tyloxapol or Tween-80, it seems unlikely that mycobacteria produce cell-associated capsules when grown planktonically. This is perhaps why the authors suggest that AM is a quorum-sensing molecule. However, quorum sensing molecules generally function when the cell density becomes high. During the lag phase, when the cell density is low, it seems rather difficult to imagine that a small amount of released AM from a few cells in the medium would have a significant role as a quorum-sensing molecule. I ask the authors to clarify their speculation on this potential role of AM.

I ran transmembrane and signal peptide prediction analyses on LamH and found that LamH has no predicted secretion signal. It has one possible transmembrane domain (aa 266-279 for Rv0365c). Where does this hydrophobic stretch locate in your predicted structure (Fig. 1b)? Can authors provide more insight on the subcellular localization of LamH and how LamH could be secreted?

TLC analyses need clarification. Indicate where the solvent front is. For example, in Fig. 2d, the authors detected PIM-like species after LamH treatment. Assuming that images show the entire TLC plates from the origin to the solvent front, it seems strange for PIM-like species to migrate near the solvent front using chloroform/methanol-based solvent. I suggest that the authors run this TLC plate side-by-side with lipid extracts containing PIMs. The identified spots from LamH-digested LM/LAM should co-migrate with AcPIM2 if their MS analysis in Fig. 2g is correct.

Regarding proteomic analysis in Fig. 4, more detailed analyses are needed to justify the claims that LAM biosynthetic enzymes are downregulated in $\Delta lamH$. How many proteins were identified in the upper-left quadrant ($P < 0.05$ and > 2 -fold change)? From the volcano plot, it appears that a thousand or more proteins are underrepresented in $\Delta lamH$. Downregulation of Aft genes may reduce arabinogalactan synthesis as well. Are galactan biosynthesis genes such as GIFT1 and GIFT2 downregulated as well? How about mycolic acid biosynthesis genes? The volcano plot seems like a massive downregulation of a large proportion of the proteome, and I wonder how specific the downregulations of these LAM biosynthetic proteins are.

Minor Comments:

Line 54: It is not clear how molecules "shaping" or "reporting" on their environmental niche is not

basic biology. The language in this sentence is also a bit vague.

Line 58: The wording at the start of the sentence suggests that the "core elements" referred to are the wide range of structures that modulate host immunity found in the mycobacterial envelope, but the rest of the sentence reads as though they refer to the core envelope components. Please clarify.

Line 61: The referenced study only describes the use of LAM in diagnostics, not LM.

Line 64: There are more than 3 structural domains mentioned: lipid anchor, mannan backbone, arabinan domain, acyl decorations, and mannose caps. While I agree that the lipid anchor, mannan backbone, and arabinan domain make up the 3 major structural domains of LAM, the text is worded such that the minor structural components are given equal weight and so it is difficult for the reader to tell what the authors mean by "three structural domains".

Line 69: Consider being more specific when mentioning the "acyl" modifications by calling them succinyl modifications.

Line 71: For general audience, please describe more precisely where the capping structures are located on the molecule.

Line 73: As far as I know, there is no way to determine the structure of LM and LAM that are only associated with the cytoplasmic membrane. Past studies suggest that LM and LAM may also associate with the outer membrane. While this remains to be proven or disproven, we cannot currently make structural statements about lipoglycans that are only associated with one of the layers of the mycobacterial envelope.

Line 84: Please elaborate more on how these references 17 and 18 support the mechanism of LM/LAM release.

Line 103: "our data provide a new perspective on LM and LAM" is vague. Please be more specific.

Line 116: The R.M.S.D. reported in the text does not match the value reported in Figure 1b.

Line 128: Mnn2 should be introduced by name here, just as Mnn1 and Mnn5 are.

Line 133: Provide more description on BT3792 so that readers can understand the function of this enzyme. I assume that it is an endo-mannosidase, but does it have any linkage specificities? Explain what makes it an "endo-like reaction profile"?

Line 137: Describe how the authors isolated mannotetraose and confirmed their structure.

Line 139: The authors concluded that "Rv0365c is a member of the GH76 family and is specific for undecorated α -1,6 mannan", but this conclusion seems premature. Specifically, the enzyme may not be specific to undecorated α -1,6 mannan as the authors' data do not exclude the possibility that other carbohydrate chains such as α -1,6 glucan are cleaved by this enzyme.

Line 185: More background information is needed to describe what the L5 site is to those who are not familiar with it.

Line 187: Both Fig. 3a and 3b should be referenced here.

Line 199: The abbreviation "2-AB" is not properly introduced.

Line 518: Provide the peak identification data as an Extended Data Figure. This is a critical piece of information to evaluate Fig. 3c.

Fig. 2c: Can the bands be labeled as Man4, Man3, etc.?

Fig. 2e: Add molecular weight markers for this and other LM/LAM SDS-PAGE.

Fig. 2f: What do error bars represent? Was the experiment done in biological replicates? Provide these details in the figure legend.

Fig. 3a-b: Are cells grown in 7H9 broth or 7H10 agar? Please clarify.

Fig. 3c-e: The descriptions in the legend for these panels are incorrect. The legend for Fig. 3e is missing.

Fig. 4a: Is the X-axis $\text{Log}_2(\text{WT}/\Delta\text{lamH})$ as depicted in the figure? Or, is it actually $\text{Log}_2(\Delta\text{lamH}/\text{WT})$ as described in the legend?

Ext Fig. 1b: Indicate which directions correspond to the first and second directions.

Ext Fig. 5: If line graph is used, the x-axis must be on an arithmetic scale. Otherwise, dot plot should be used.

Supplemental Table 1: Provide gene names in the column AH. It is difficult for readers to navigate the table without having Rv numbers.

Reviewer #2 (Remarks to the Author):

This paper identifies and characterizes an enzyme that cleaves arabinomannans off of LM and LAM, two important surface glycolipids in mycobacteria. The authors then characterize mutants of the identified proteins, LamH, in Msmeg, BCG and Mtb. They show the importance of this enzyme is exiting lag phase in axenic growth, and in survival in macrophages. The data provided are clear and the experiments have been done rigorously. The conclusions are supported by the data. This work is important to the field of mycobacterial cell surface metabolism, and to our understanding of the host-pathogen interface in Mtb infections. My only comments are minor.

Enzymatic assays show that LamH can clear short a-1,6 mannans without any a-1,2 decorations.

Panel 2d – please indicate which of these TLCs was stained with which method.

Line 175 – the proper annotation for a transposon mutant is $\text{lamH}::\text{tn5}$, if it is a tn5 transposon. Also, fig 3 and text: the proper annotation for a complementation vector at the L5 site would be $\text{L5}::\text{lamH}$. Your notation of $\Delta\text{lamH}::\text{lamH}$ means that you have a full deletion of lamH , and it is complemented through an integration at the lamH locus. The correct annotation for the complementation strain, assuming a tn5 tn, is: $\text{lamH}::\text{tn5}$ $\text{L5}::\text{lamH}$.

Figure 3 - I think the legend is mixed up here in places. The description for panel B seems to be about capsule, but the data suggest it is about lipid-anchored LM. The description for panel C is incomplete – please describe the whole figure panel and where it comes from. In general there are more methodological details than is typical in this legend. I think it is better to focus on just describing the data. The panel D legend also doesn't seem to match the data, and the panel E legend is missing.

Fig 5b – can you clarify which section of the curve this refers to? Is this lag phase? Please indicate which time points were included for each sample.

Lines 236-242 – this is confusing. What part of a growth curve is linear? I would guess the lag phase, since that is what you are talking about, but then you say the exponential growth rate is reduced. Exponential growth is, by definition, exponential, not linear. If you want to compare the

exponential parts of the curve, you need to plot your data on a log₂ scale, and then find the linear part of THAT curve, is that what you did? Once you have found the exponential (ie, not lag or stat phase) parts of your growth curve, you should then fit that part of the curve to an exponential growth equation to find the rate or doubling time of growth – you should definitely not fit it to a linear equation. Your curves in A and C are on a linear scale, and you don't mention putting your data on a log₂ scale so it's not clear what you've done.

Extended data 5 – you did not integrate the lamH gene into pLJR965, did you? I think you mean: pLJR965- sgRNA(lamH). And, in Fig 6, similarly, did you integrate an unmodified crispr vector into the lamH locus? That is what your annotation suggests. I think the strain you're using is: H37Rv L5::pLJR965 - sgRNA(lamH)

Line 272 –Your data suggests that LamH is important for survival in macrophages, but survival is not equivalent to pathogenesis.

Methods: please provide methods about how you did the orcinol or MPA staining, or provide a citation of a paper where this method is described.

Reviewer #3 (Remarks to the Author):

In this manuscript, Franklin et al. show that a putative mycobacterial glycosidase (Rv065c in *M. tuberculosis* or Mtb) that they name LamH has activity consistent with cleaving the extended carbohydrate components from the abundant lipoglycans lipomannan (LM) and lipoarabinomannan (LAM). Using structural homology analyses followed by admirably extensive and thorough in vitro and cellular assays in several mycobacterial species (Mtb, BCG, *M. smegmatis*), they find that LamH specifically cleaves α -1,6-mannoside linkages and that loss of function in lamH correlates with a reduction in detectable capsular arabinomannan. Interestingly, loss of lamH also delays the transition from lag to exponential phase growth– and this can be at least partially rescued by exogenous supplementation of LAM components!-- and attenuates Mtb replication in infected macrophages. These observations support an intriguing role for LamH cleavage products such as arabinomannan in enabling growth and survival of Mtb. The reported data are generally sound and the broad range of assays support the conclusions, with several exceptions (see further below). More broadly, the authors do not address the discrepancy between the proposed involvement of LamH in releasing arabinomannan from LM/LAM that is destined for the capsule and the absence of any export signal on LamH (i.e., its presumed cytoplasmic localization). Pending addressing these concerns, the reported results constitute a sound and original contribution to the field and therefore appropriate for publication in Nat Comm.

Detailed comments

MAIN TEXT:

P.1, line 45: For ease of reference for readers, we suggest including the Mtb identifier, e.g., "that we term LamH (Rv0365c in *Mycobacterium tuberculosis*) if space permits.

Throughout, please carefully review the correct use of "which, the occurrence of misplaced modifiers, and sentence structure in general to ensure clear and correct expression of the desired meaning (also please proofread for typographical errors). One example of many:

P.1, lines 43-44: "An important class of exported LM/LAM is the capsular derivative of these molecules, which is devoid of its lipid anchor" – "it" is LM/LAM, but "which" currently modifies "molecules" and this makes the overall parsing of the sentence and therefore the meaning difficult to follow. Please reword this and other analogous instances.

P.3, line 58: "The core elements of this are..." Would suggest tempering wording to "the core elements include", as citing only mAGP and lipoglycans ignores many acknowledged major lipid components with important roles in host-pathogen interactions such as TDM, TMM, etc.

P.3, line 71: "the precise structure of LM and LAM associated with the cytoplasmic membrane can vary" We are not aware of any data discriminating the structure of LM/LAM associated with the plasma membrane vs. other components of the cell envelope. Please clarify or remove the phrase "associated with the cytoplasmic membrane".

P.4, subject-verb agreement: change

Line 79: "pathways... is" to "pathways... are"

Line 99: "data also shows" to "data also show"

P.5, line 100: "AM can act as a signaling molecule whose accumulation helps the bacteria" - We suggest tempering this claim, e.g., "capsular AM aids the transition to exponential growth" as better reflecting experiments that show how reduction in capsular AM and addition of exogenous AM affects growth kinetics.

P. 5 line 102: Similarly, we suggest tempering the claim that LamH knockdown results in a "decrease in pathogenesis" in favor of reporting the findings as is: reduced survival in macrophages.

P. 6 line 128: "strains lacking glycosyltransferases that produce three distinct mannans of decreasing complexity" - The parsing of this sentence is unclear, please reword for clarity. Furthermore and also for clarity, in this section, please more clearly define the nomenclature (Mnn1, 2, and 5 and that these are names of yeast strains?) to match what can currently be discerned/inferred only from the figure.

P.7, line 150: Please define what is detected by the ProQ-Emerald assay, rather than just name it.

P.7, line 156: What is meant by the additional information "numbering relative... to genomic annotation" is not clear. It is generally assumed that numbering is derived from ORF annotation and it is not necessary to make a note unless there is some discrepancy between strains or reported genomes. Please clarify or remove this parenthetical phrase.

P.7, line 166: "we next sought to identify the limit glycolipid product" Is "limiting" meant? Please clarify/correct.

P.8, line 176: Please add citation for "99% genetically identical"

Line 177: We suggest tempering the subjective "excellent" with the more circumspect "well-accepted".

P. 9 lines 201-203: "This analysis indicated that the complemented mutant produced significantly less AM than the wild type (Fig. 3d), showing that complementation is imperfect, and suggesting that expression of the lamH gene is altered when located distally." This suggests that there is reduced expression of lamH in the complement strain, but the complemented gene is in the pMV306 plasmid driven by the Hsp60 promoter and therefore would likely be overexpressed. Please provide a different potential explanation for the reduced AM levels, or use RT-qPCR to demonstrate that there is truly lower expression of the lamH gene.

P. 9 line 209: We suggest tempering the claim that "increased abundance of LM/LAM triggers a membrane stress response."

P.10, line 224: consistent verb tense, change "are" to "were"

Line 225: "this provides evidence that levels of LM/LAM are tightly regulated" We suggest tempering the language by removing "tightly" unless there is specific evidence to support this qualification; the point is otherwise still significant, that there regulation in the balance of intermediates/products in this pathway at all.

P.10/11, all data regarding the analysis of exponential growth rates (lines 236-240, lines 246-252, figure 5b,d). These arguments are quite complicated, and indeed for the rescue experiments the observations become less consistent (partial rescue of lag time, but decrease in growth rate). More

to the point, the changes (up to 20%) in growth rate, although statistically significant, are not clearly biologically meaningful. Overall, we see these as comparatively weak arguments that are not necessary for the main conclusion that LamH activity is required for efficient exit from lag-phase growth and recommend that they be removed.

P.12, line 270: Significant figures. Unless the authors can show that a 1.01-log reduction is reproducible, we recommend that they report a smaller number of significant figures (e.g., 0.4 and 1). More importantly, Figure 6 is reported as “data representative of two independent experiments with similar results”. If this is the case, then how many replicates are represented in the figure to support the statistical analysis? What statistical analysis was performed?

P. 13 lines 301-311: The three hypotheses for LM and LAM release to the host do not make sense as distinct possibilities. The mention of LprG likely only accounts for transport across the inner membrane, unless the authors are invoking a hypothesis through which LM and LAM are somehow secreted bound to LprG. The mentions of vesicle production and the presence of host antibodies against capsular AM do not sufficiently explain release of LM and LAM. Moreover, AM is a distinct species from LM/LAM and its presence in the capsule does not explain how LM/LAM reach host compartments. Please clarify the ideas in this section.

Also, within this section, according to the literature, LprG has the capacity to bind to a variety of diacyl/triacyl lipids including LM/LAM and triacylglycerides, so roles in the transport of both classes are not necessarily mutually exclusive. We suggest rewording.

Most crucially, regarding the key statement, “this work provides the first mechanistic basis for the generation of capsular AM”, overlooks the fact that LamH does not appear to have a predicted canonical secretion signal (or any N-terminal features consistent with one, such as even a weak predicted TM helix). Its presumed location in the cytoplasm is then inconsistent with a role in generating AM from LM/LAM in the periplasm. Moreover, the model for LAM/LM synthesis shown in Fig. 1 and 3 is not fully consistent with commonly reported models. We recommend that the authors more clearly depict the ambiguity in the identity and localization of the GTs that create AcPIM3/4 from AcPIM2. Indeed, most models depict AcPIM4 as the substrate of a presumed flippase that moves this species into the periplasm, consistent with the apparent bifurcation between further elaboration into AcPIM5/6 vs. LAM/LM, both by periplasmic GTs. If we instead presume that AcPIM4 is first made in the cytoplasm, then cytoplasmic LamH could be involved in regulating the relative pools of AcPIM1 and AcPIM4, as a means of regulating the metabolic flux into downstream pathways. Admittedly, while this is consistent with the increase in LAM/LM in the lamH mutant, it does not easily explain the concomitant reduction in AM. Nevertheless, the authors should address the issue of enzyme localization in reconciling their model for the physiological role of LamH..

METHODS:

P. 15: Please include methods for bioinformatic protein analyses.

P. 33-34: Please include the enzyme concentration and the solvent system also in the figure legends throughout the manuscript. Also, please report the number of biological replicates performed even if only showing representative data.

Line 347/349: Reported culture temperatures are 37C for yeast and 30C for *M. smegmatis*– is this correct? It seems the reverse of the typical growth temps for these microbes.

Line 364: Please clarify this section and heading to reflect that this mutagenesis is to generate a lamH KO. Please also include the MSMEG gene identifier here and the oligonucleotide sequence (there is no need for a supplemental table for a single oligo).

P. 16, line 378: “nearby genes” Please be more specific about which genes and also report the primer sequences used. More generally, we recommend that in reporting this analysis, the authors describe the context for lamH. Is it a single-gene operon? Do the surrounding genes have known or predicted functions that could be related? Did the authors also do a synteny analysis and see

any conservation of the gene context?

P. 20, line 501: Especially given that this is not necessarily a widely used protocol, please provide the resuspension volume and provide a more reproducible direction than "gently vortexed" – e.g., vortexed at the lowest setting?

P.21, line 522: please be more specific regarding the "portable spectrophotometer" if this is somehow critical to how the measurements were made. Else, monitoring of OD600 is considered standard and this vague specification should be removed for clarity.

P. 21, line 530: Please provide the methods for the preparation of the "acetone-precipitated proteome samples".

FIGURES:

Figure 1. We appreciate the authors' effort to depict the glycosidic bond connectivities, but the similar colors used for both sugars and the very short bonds makes these differences indiscernible. Also, the caption states "Grey text indicates glycolipid species", but grey text appears instead to name the GTs? Please amend for clarity/correctness.

Figure 2. As noted above, please report the number of independent experiments that these data represent, for each experiment.

Figure 3. The caption and numbering appears to be off– i.e., a/b used to be combined LAM/LM analysis, but now are separate. Also, "Peaks were identified by digestion with appropriate enzymes" is vague and these methods do not appear to be provided in the methods section. Please amend.

Figure 5. Please clarify the legends vs. the caption. The legends between a), which refers to strains, and b), which refers to type of spent medium, is confusing. Also, the legend states "+AM", consistent with this being a LamH-digested preparation of LAM, but the caption says "0.5 mg/mL LAM". Please amend.

Figure 6. Please see comments above regarding # of independent replicates shown and subjected to statistical analysis, given that the caption states that data are representative of 2 experiments.

Reviewer #4 (Remarks to the Author):

My comments were combined with another reviewer's

RESPONSE TO REVIEWER COMMENTS

We have extensively revised the manuscript in response to the Reviewers' comments. To facilitate review, we have tracked the changes in the manuscript made in response to the specific concerns of the Reviewers outlined below.

Reviewer #1 (Remarks to the Author):

This study reports the identification of LamH, a mannosidase that releases arabinomannan from LAM. The presence of arabinomannan and mannan in the mycobacterial capsule has been known for a long time, but it remained unknown if there is an enzyme that actively engages in hydrolyzing LAM to release these capsular glycans. This study elegantly fills in this gap using bioinformatic, genetic, and biochemical approaches. It is a well-written manuscript, and the study is of high significance to the field of mycobacteriology and glycobiology.

We thank the reviewer for their time and appreciate their enthusiasm for the work.

Below, I have several comments to improve the clarity of the manuscript.

Major Comments:

1. The AM acting as a signalling molecule is a farfetched concept. AM may be acting simply as an absorbent to remove a molecule that inhibits the exit of cells from lag phase. Such an effect of AM would be considered an indirect effect of accumulating AM. I suggest the authors to provide more solid evidence to prove the role of AM as a signalling molecule, or tone down their claims.

We recognise that we have perhaps overstated our results regarding signalling. In response, we have toned down the language in the manuscript pertaining to these results. Please see in particular lines 48, 99, 262, 358-64. While work is underway to delineate the potential signalling pathway, we feel that waiting for those results will unnecessarily delay this manuscript, which focuses on the LamH-catalysed release of arabinomannan. The reviewer's suggestion of AM acting to absorb some other molecule would not remove it from this putative pathway but instead change its role. We are open to and appreciate them sharing this hypothesis and look forward to finding the answer in future work.

2. The growth conditions are unclear. For some experiments, authors grew cells on 7H10 agar plate and extracted AM from cells by vortexing. However, for experiments in Fig. 5 and Extended Data Fig. 3, the cells were grown in 7H9 broth planktonically. Does 7H9 broth contain ADC and 0.05% tyloxapol (like the authors did to grow Mtb)? If *M. smegmatis* was grown in a complete 7H9 medium, why does the culture go into stationary phase at the OD600 of ~1.2 in Extended Data Fig. 3d? In 7H9, *M. smegmatis* can normally reach to OD600 = 4~5.

To extract capsular polysaccharides, cells were grown on 7H10 agar plates at 37 °C with 5% CO₂. For growth curve cultures, cells were grown static in 7H9 media supplemented with ADC at 37 °C with 5% CO₂. The 7H9 media used to culture *M. bovis* BCG and *M. smegmatis* did not contain 0.05% tyloxapol; we used 0.05% Tween

80 instead. The OD_{600nm} the culture reaches can vary with growth conditions and the spectrophotometer used to measure the culture. We have found in our laboratory, under the growth conditions tested with the small OD reader we use (a portable unit that can go into biosafety cabinets for Biosafety Level-2 work), *M. smegmatis* reproducibly reaches an OD_{600nm} maximum of around 1.2. However, in our view, the final OD_{600nm} of the culture is not important to the outcome of the experiment but rather the shape and reproducibility of the growth curve(s).

3. Assuming that the culture medium contained either tyloxapol or Tween-80, it seems unlikely that mycobacteria produce cell-associated capsules when grown planktonically. This is perhaps why the authors suggest that AM is a quorum-sensing molecule. However, quorum sensing molecules generally function when the cell density becomes high. During the lag phase, when the cell density is low, it seems rather difficult to imagine that a small amount of released AM from a few cells in the medium would have a significant role as a quorum-sensing molecule. I ask the authors to clarify their speculation on this potential role of AM.

While we understand the Reviewer's concern, the term 'quorum' as we understand it means a minimum number is present. That may be a relatively large number of cells, as is in the case of classical *Pseudomonas* quorum-sensing, but the precise number of cells required to be considered a 'quorum' will depend on the biological system being studied. Phrased differently, the number of cells required to reach a quorum depends on the sensitivity of the quorum-sensing system. We hypothesise that the consistent action of LamH will result in a gradual increase in AM concentration in the growth media, and as the reviewer rightly points out, the non-covalent nature of the capsule means that this material will be shed into the surrounding environment. Our data suggest, but do not prove, that at some critical AM concentration, the bacteria transition to exponential growth. Nonetheless, having considered the comments from the Reviewers and our data, we recognise that the language in the manuscript should be toned down in this regard. To not mislead the reader or overstate the data, we have rephrased these statements accordingly. Please see our response to the Reviewer's first comment for the edited lines.

4. I ran transmembrane and signal peptide prediction analyses on LamH and found that LamH has no predicted secretion signal. It has one possible transmembrane domain (aa 266-279 for Rv0365c). Where does this hydrophobic stretch locate in your predicted structure (Fig. 1b)? Can authors provide more insight on the subcellular localization of LamH and how LamH could be secreted?

This point was also raised by Reviewers 2 and 3. Despite the apparent lack of a signal peptide on LamH, five independent proteomics studies have placed the enzyme in the membrane or cell envelope fractions in *M. tuberculosis* and *M. smegmatis*¹⁻⁵. We have added the references cited here and a new sentence at lines 115-117 in the manuscript to reflect this. There is also precedence in the mycobacterial literature for proteins with no discernible signal peptide being secreted, as shown by the group of Prof. Miriam Braunstein⁶. So, while we initially shared the Reviewer's concerns about LamH localisation, all available data would support the conclusion that it is associated with the membrane and processes LM/LAM in this context.

5. TLC analyses need clarification. Indicate where the solvent front is. For example, in Fig. 2d, the authors detected PIM-like species after LamH treatment. Assuming that images show the entire TLC plates from the origin to the solvent front, it seems strange for PIM-like species to migrate near the solvent front using chloroform/methanol-based solvent. I suggest that the authors run this TLC plate side-by-side with lipid extracts containing PIMs. The identified spots from LamH-digested LM/LAM should co-migrate with AcPIM2 if their MS analysis in Fig. 2g is correct.

The figures have been edited to clarify the location of the solvent front. We had initially truncated the TLC to show just the bands of interest, but now show the origin to the solvent front for all TLCs (Figures 2a,b,c,d,e). Per journal requirements, we now present the entire unedited TLC as imaged (along with all other TLCs and gels) as a new supplemental Figure 8. We have also taken the Reviewer's excellent suggestion and re-run the reaction products on a TLC system comprised of chloroform:methanol:13 M ammonia:1 M ammonium acetate:water (180: 140:9:9:23 v/v/v/v/v) which is derived from PMID 35952902⁷. This includes purified PIMs, the LamH product, and a co-spot of both samples. This is now Figure 2g, which shows that the reaction product co-migrates with AcPIM₂.

Regarding proteomic analysis in Fig. 4, more detailed analyses are needed to justify the claims that LAM biosynthetic enzymes are downregulated in $\Delta lamH$. How many proteins were identified in the upper-left quadrant ($P < 0.05$ and >2 -fold change)? From the volcano plot, it appears that a thousand or more proteins are underrepresented in $\Delta lamH$.

The reviewer is correct in that within $\Delta lamH$, many of the proteome changes observed correspond to reductions in protein abundances. Based on a threshold of $P < 0.01$ and $>\pm 1$ -fold change, we observe 215 proteins decreasing in abundance and 29 increased within $\Delta lamH$ out of the total 1916 proteins identified (see lines 228-232). It is important to note that while loss of *lamH* does drive large proteome alterations, much of the proteome is unaffected, as demonstrated from assessments of the observed MaxLFQ values of WT / $\Delta lamH$ replicates with comparable distributions of protein abundances observed (See Supplementary Fig. 6).

Downregulation of Aft genes may reduce arabinogalactan synthesis as well. Are galactan biosynthesis genes such as GltT1 and GltT2 downregulated as well? How about mycolic acid biosynthesis genes? The volcano plot seems like a massive downregulation of a large proportion of the proteome, and I wonder how specific the downregulations of these LAM biosynthetic proteins are.

Neither GltT1(FCU26_3927/Rv3782) nor GltT2 (FCU26_3953/Rv3808c) is significantly altered in abundance compared to the wild type. We see altered abundance in enzymes that are likely involved in lipid metabolism (several FadD and FadE-type enzymes for example), but key mycolic acid-associated proteins such as the antigen 85 complex (Rv3804c, Rv1886c, Rv0129c), MMPL3 (Rv0206c) and Pks13 (Rv3800c) are not significantly different in abundance in our dataset. However, in response to the reviewer's question, we have undertaken a Gene Ontology analysis of the proteomics data. To enable this, we have used the PATRIC

database to match the observed *M. bovis* BCG proteome to the *M. tuberculosis* H37Rv genome and conducted an enrichment analysis of proteome changes using Fisher exact tests within Perseus using a 5% FDR and a Benjamini-Hochberg multiple hypothesis correction. This revealed a significant enrichment in GO terms for proteins associated with 'plasma membrane' and 'peptidoglycan-based cell wall'. These data are consistent with but do not prove, that there is a membrane stress response in response to increased LAM in the cytoplasmic membrane. Our observation of increased acylation of PIMs (increase in Ac₂PIM₂), which was previously associated with membrane stress, also supports this hypothesis. This will merit further study but is beyond the scope of this manuscript. There is also a small but significant anti-correlation for the terms 'translation' and structural constituent of the ribosome' (Line 244- 251, Supplementary Fig. 6).

Minor Comments:

Line 54: It is not clear how molecules "shaping" or "reporting" on their environmental niche is not basic biology. The language in this sentence is also a bit vague.

We have edited this section to be more specific. Please see lines 52-59.

Line 58: The wording at the start of the sentence suggests that the "core elements" referred to are the wide range of structures that modulate host immunity found in the mycobacterial envelope, but the rest of the sentence reads as though they refer to the core envelope components. Please clarify.

We have reworded for clarity (lines 54-59).

Line 61: The referenced study only describes the use of LAM in diagnostics, not LM.

Our apologies, we have corrected the statement to reflect this (line 61).

Line 64: There are more than 3 structural domains mentioned: lipid anchor, mannan backbone, arabinan domain, acyl decorations, and mannose caps. While I agree that the lipid anchor, mannan backbone, and arabinan domain make up the 3 major structural domains of LAM, the text is worded such that the minor structural components are given equal weight and so it is difficult for the reader to tell what the authors mean by "three structural domains".

We have re-worded this sentence to be more clear, drawing a distinction between 3 major structural domains and minor modifications (lines 63-73).

Line 69: Consider being more specific when mentioning the "acyl" modifications by calling them succinyl modifications.

We have reworded accordingly (line 69).

Line 71: For general audience, please describe more precisely where the capping structures are located on the molecule.

We have added a line to reflect their location at the termini of the D-arabinan domain (line 71).

Line 73: As far as I know, there is no way to determine the structure of LM and LAM that are only associated with the cytoplasmic membrane. Past studies suggest that LM and LAM may also associate with the outer membrane. While this remains to be proven or disproven, we cannot currently make structural statements about lipoglycans that are only associated with one of the layers of the mycobacterial envelope.

This line was meant to distinguish LAM from capsular derivatives (rather than possible OM-associated derivatives), but we recognise that the text is unclear. We have edited this line to reflect the current state of understanding (lines 74-76).

Line 84: Please elaborate more on how these references 17 and 18 support the mechanism of LM/LAM release.

We were using the release of peptidoglycan as an analogy for how cell wall components are shed by bacteria. We have edited the sentence to clarify this point (lines 83-84).

Line 103: “our data provide a new perspective on LM and LAM” is vague. Please be more specific.

This has been removed from the manuscript (line 101).

Line 116: The R.M.S.D. reported in the text does not match the value reported in Figure 1b.

Apologies, this was initially measured using the ‘Matchmaker’ function in ChimeraX, but upon colleague feedback, we decided to use the more accurate tools found in Coot (LSQKab). We neglected to update Figure 1 correctly. This has been corrected in Figure 1b.

Line 128: Mnn2 should be introduced by name here, just as Mnn1 and Mnn5 are.

We have edited this as requested (lines 126-130).

Line 133: Provide more description on BT3792 so that readers can understand the function of this enzyme. I assume that it is an endo-mannosidase, but does it have any linkage specificities? Explain what makes it an “endo-like reaction profile”?

Thank you for the suggestion. We have added text describing BT3792 and its function/substrate specificity (Lines 133-135). By endo-like profile, we mean that if the enzyme was exo-acting, we would expect the major product to be mannose after incubation of that timescale (18 h). Instead, we see a series of oligosaccharide sizes suggestive of the enzyme cutting within the glycan. This is not in and of itself conclusive of endo-activity, which is why we use the term ‘endo-like’. The ability to digest LM/LAM, however, strongly supports the designation of the enzyme as endo-acting. We have edited this section for clarity (lines 135-143).

Line 137: Describe how the authors isolated mannotetraose and confirmed their structure.

The mannotetraose was purified using established protocols outlined in Cuskin, Lowe et al, 2015. We have updated the methods section to include more detail about this. Furthermore, in the time since the manuscript was submitted, we have gained access to an IC-MS system and so have taken that material and re-confirmed that its molecular weight is consistent with mannotetraose. These data are presented in Supplemental Figures 2b and 2c. We have also updated the text at line 145.

Line 139: The authors concluded that “Rv0365c is a member of the GH76 family and is specific for undecorated α -1,6 mannan”, but this conclusion seems premature. Specifically, the enzyme may not be specific to undecorated α -1,6 mannan as the authors’ data do not exclude the possibility that other carbohydrate chains such as α -1,6 glucan are cleaved by this enzyme.

While we have not conducted an exhaustive test of substrates, we did test the enzyme against capsular alpha-glucan (see new Supplemental Fig. 2c) which showed no apparent activity. Nevertheless, we have tempered this statement in lines 146-148.

Line 185: More background information is needed to describe what the L5 site is to those who are not familiar with it.

We have edited the text to clarify this (line 197).

Line 187: Both Fig. 3a and 3b should be referenced here.

We have edited the text to reflect this (line 199).

Line 199: The abbreviation “2-AB” is not properly introduced.

We have edited the text to reflect this (line 207).

Line 518: Provide the peak identification data as an Extended Data Figure. This is a critical piece of information to evaluate Fig. 3c.

We have included this now as a new Supplementary Table 2. Compiling this table led us to recognise some labelling errors in the figure, which we have corrected now (Figure 2h). We apologise for the oversight.

Fig. 2c: Can the bands be labeled as Man4, Man3, etc.?

This has now been annotated (Figure 2c).

Fig. 2e: Add molecular weight markers for this and other LM/LAM SDS-PAGE.

This has now been corrected for Figure 2e. However, the LAM gels in the supplemental materials were run with a ladder that does not show up in the ProQ

Emerald Staining procedure. We have, however, provided a gel with one of each of the samples run with a new ladder that does show up during this staining procedure to confirm they migrate at the expected sizes. This is included with a new figure showing the complete, uncropped TLCs and gels for all figures (Supplementary Fig. 8).

Fig. 2f: What do error bars represent? Was the experiment done in biological replicates? Provide these details in the figure legend.

Apologies, these are standard deviations of 3 biological replicates. This has now been provided (Figure 2f).

Fig. 3a-b: Are cells grown in 7H9 broth or 7H10 agar? Please clarify.

This has been clarified.

Fig. 3c-e: The descriptions in the legend for these panels are incorrect. The legend for Fig. 3e is missing.

Apologies, this has been corrected.

Fig. 4a: Is the X-axis $\text{Log}_2(\text{WT}/\Delta\text{lamH})$ as depicted in the figure? Or, is it actually $\text{Log}_2(\Delta\text{lamH}/\text{WT})$ as described in the legend?

The figure is correct; the legend has been edited to clarify this.

Ext Fig. 1b: Indicate which directions correspond to the first and second directions.

The running directions have been indicated in the figure.

Ext Fig. 5: If line graph is used, the x-axis must be on an arithmetic scale. Otherwise, dot plot should be used.

We have re-plotted the data as a mixed bar/dot plot on a semi-log scale (Supplementary Figure 7).

Supplemental Table 1: Provide gene names in the column AH. It is difficult for readers to navigate the table without having Rv numbers.

We have added this column to the data.

Reviewer #2 (Remarks to the Author):

This paper identifies and characterizes an enzyme that cleaves arabinomannans off of LM and LAM, two important surface glycolipids in mycobacteria. The authors then characterize mutants of the identified proteins, LamH, in Msmeg, BCG and Mtb. They show the importance of this enzyme is exiting lag phase in axenic growth, and in survival in macrophages. The data provided are clear and the experiments have been done rigorously. The conclusions are supported by the data. This work is important to the field of mycobacterial cell surface metabolism, and to our

understanding of the host-pathogen interface in Mtb infections. My only comments are minor.

We thank the reviewer for their positive comments and appreciate their assessment of the manuscript.

Enzymatic assays show that LamH can clear short a-1,6 mannans without any a-1,2 decorations.

Panel 2d – please indicate which of these TLCs was stained with which method.

This has now been amended in the figure legend and on the figure by labelling ‘lipid’ or ‘carbohydrate’. The original TLC was false-coloured for presentation, but we felt this may confuse more experienced users (MPA is normally green, not blue), so we have reverted to black-and-white colouring where the original images were recorded in black-and-white. We have also added the TLC of the catalytic null reaction for completeness (Figure 2d).

Line 175 – the proper annotation for a transposon mutant is *lamH::tn5*, if it is a *tn5* transposon. Also, fig 3 and text: the proper annotation for a complementation vector at the L5 site would be *L5::lamH*. Your notation of $\Delta lamH::lamH$ means that you have a full deletion of *lamH*, and it is complemented through an integration at the *lamH* locus. The correct annotation for the complementation strain, assuming a *tn5* *tn*, is: *lamH::tn5 L5::lamH*.

We apologise for this oversight. We have now edited the terminology throughout. When we first introduce the strain, to reduce complexity later on we identify $\Delta lamH$ as referring to *lamH::Himar1*. The complementation is not at the original locus. This is from an integrative plasmid at the L5 attP site in the genome using the vector pMV306. This results in the strains $\Delta lamH L5::pMV306-lamH$ and $\Delta lamH L5::pMV306-lamH_{D96A}$. Importantly, this vector does not include a constitutive promoter, so any expression of *lamH* is a consequence of the 300 bp 5’ of the gene we included in our construct to capture possible promoter elements. This additional 300 bp segment does not include any predicted open reading frames. Please see our response to Reviewer 3 below and the edited text line 196-199 about the complementation for further details. We have also updated the legend for Figures 3 and 5.

Figure 3 - I think the legend is mixed up here in places. The description for panel B seems to be about capsule, but the data suggest it is about lipid-anchored LM. The description for panel C is incomplete – please describe the whole figure panel and where it comes from. In general there are more methodological details than is typical in this legend. I think it is better to focus on just describing the data. The panel D legend also doesn’t seem to match the data, and the panel E legend is missing.

Thank you for identifying this. We have edited this legend to reflect your and the other Reviewer’s comments and can only apologise for the errors. We have also removed some methodological information from the legend.

Fib 5b – can you clarify which section of the curve this refers to? Is this lag phase? Please indicate which time points were included for each sample.

Please see below.

Lines 236-242 – this is confusing. What part of a growth curve is linear? I would guess the lag phase, since that is what you are talking about, but then you say the exponential growth rate is reduced. Exponential growth is, by definition, exponential, not linear. If you want to compare the exponential parts of the curve, you need to plot your data on a log₂ scale, and then find the linear part of THAT curve, is that what you did? Once you have found the exponential (ie, not lag or stat phase) parts of your growth curve, you should then fit that part of the curve to an exponential growth equation to find the rate or doubling time of growth – you should definitely not fit it to a linear equation. Your curves in A and C are on a linear scale, and you don't mention putting your data on a log₂ scale so it's not clear what you've done.

You are correct, and we regret that we initially analysed these data incorrectly. We have removed this analysis in response to your comments and, as suggested by Reviewer 3, because they do not contribute to the paper's conclusions.

Extended data 5 – you did not integrate the lamH gene into pLJR965, did you? I think you mean: pLJR965- sgRNA(lamH). And, in Fig 6, similarly, did you integrate an unmodified crispr vector into the lamH locus? That is what your annotation suggests. I think the strain you're using is: H37Rv L5::pLJR965 - sgRNA(lamH)

We have edited the text to reflect the reviewer's helpful suggestion. Our apologies for the nomenclature errors (Supplementary Figure 7).

Line 272 –Your data suggests that LamH is important for survival in macrophages, but survival is not equivalent to pathogenesis.

The text has been edited to reflect this. Please see lines 103 and 296 for amended text.

Methods: please provide methods about how you did the orcinol or MPA staining, or provide a citation of a paper where this method is described.

These have now been added to the methods section (lines 478-483).

Reviewer #3 (Remarks to the Author):

In this manuscript, Franklin et al. show that a putative mycobacterial glycosidase (Rv065c in *M. tuberculosis* or Mtb) that they name LamH has activity consistent with cleaving the extended carbohydrate components from the abundant lipoglycans lipomannan (LM) and lipoarabinomannan (LAM). Using structural homology analyses followed by admirably extensive and thorough in vitro and cellular assays in several mycobacterial species (Mtb, BCG, *M. smegmatis*), they find that LamH specifically cleaves α -1,6-mannoside linkages and that loss of function in lamH correlates with a

reduction in detectable capsular arabinomannan. Interestingly, loss of lamH also delays the transition from lag to exponential phase growth— and this can be at least partially rescued by exogenous supplementation of LAM components!— and attenuates Mtb replication in infected macrophages. These observations support an intriguing role for LamH cleavage products such as arabinomannan in enabling growth and survival of Mtb. The reported data are generally sound and the broad range of assays support the conclusions, with several exceptions (see further below). More broadly, the authors do not address the discrepancy between the proposed involvement of LamH in releasing arabinomannan from LM/LAM that is destined for the capsule and the absence of any export signal on LamH (i.e., its presumed cytoplasmic localization). Pending addressing these concerns, the reported results constitute a sound and original contribution to the field and therefore appropriate for publication in Nat Comm.

We thank the Reviewer for considering the manuscript and have addressed their comments below.

Detailed comments
MAIN TEXT:

P.1, line 45: For ease of reference for readers, we suggest including the Mtb identifier, e.g., “that we term LamH (Rv0365c in Mycobacterium tuberculosis) if space permits.

Thank-you for this suggestion, we have amended it accordingly (Lines 44 and 93).

Throughout, please carefully review the correct use of “which, the occurrence of misplaced modifiers, and sentence structure in general to ensure clear and correct expression of the desired meaning (also please proofread for typographical errors). One example of many:

We thank the reviewer for their helpful suggestions on the text. We have now re-edited the examples below and others that we were able to locate. Where substantial changes were made, this has been tracked throughout the document.

P.1, lines 43-44: “An important class of exported LM/LAM is the capsular derivative of these molecules, which is devoid of its lipid anchor” – “it” is LM/LAM, but “which” currently modifies “molecules” and this makes the overall parsing of the sentence and therefore the meaning difficult to follow. Please reword this and other analogous instances.

We have edited this (lines 41-43).

P.3, line 58: “The core elements of this are...” Would suggest tempering wording to “the core elements include”, as citing only mAGP and lipoglycans ignores many acknowledged major lipid components with important roles in host-pathogen interactions such as TDM, TMM, etc.

We have edited the text accordingly (lines 52-59).

P.3, line 71: “the precise structure of LM and LAM associated with the cytoplasmic membrane can vary” We are not aware of any data discriminating the structure of LM/LAM associated with the plasma membrane vs. other components of the cell envelope. Please clarify or remove the phrase “associated with the cytoplasmic membrane”.

We have edited the text accordingly (lines 74-76).

P.4, subject-verb agreement: change

Line 79: “pathways... is” to “pathways... are”

Corrected (line 80).

Line 99: “data also shows” to “data also show”

Corrected (line 98)

P.5, line 100: “AM can act as a signaling molecule whose accumulation helps the bacteria” - We suggest tempering this claim, e.g., “capsular AM aids the transition to exponential growth” as better reflecting experiments that show how reduction in capsular AM and addition of exogenous AM affects growth kinetics.

We thank the reviewer for the suggestion and have tempered the claim accordingly. (lines 99)

P. 5 line 102: Similarly, we suggest tempering the claim that LamH knockdown results in a “decrease in pathogenesis” in favor of reporting the findings as is: reduced survival in macrophages.

As mentioned in response to another Reviewer, we have tempered the claims in this sentence (line 101).

P. 6 line 128: “strains lacking glycosyltransferases that produce three distinct mannans of decreasing complexity” - The parsing of this sentence is unclear, please reword for clarity. Furthermore and also for clarity, in this section, please more clearly define the nomenclature (Mnn1, 2, and 5 and that these are names of yeast strains?) to match what can currently be discerned/inferred only from the figure.

We have rewritten this section to better explain the source/nature of these strains and the nomenclature. (lines 125-130)

P.7, line 150: Please define what is detected by the ProQ-Emerald assay, rather than just name it.

We have added a line explaining the function of ProQ-Emerald (lines 158-160).

P.7, line 156: What is meant by the additional information “numbering relative... to genomic annotation” is not clear. It is generally assumed that numbering is derived from ORF annotation and it is not necessary to make a note unless there is some discrepancy between strains or reported genomes. Please clarify or remove this

parenthetical phrase.

It is reasonably common to see in papers amino acid numbering that corresponds to expression constructs rather than genomic constructs. This often creates confusion when analysing protein structures or comparing protein sequences. We included this statement to remove any doubt but have removed it in response to the Reviewer's suggestion (line 164).

P.7, line 166: "we next sought to identify the limit glycolipid product" Is "limiting" meant? Please clarify/correct.

We have edited the text for clarity. (Line 171-178)

P.8, line 176: Please add citation for "99% genetically identical"

Line 177: We suggest tempering the subjective "excellent" with the more circumspect "well-accepted".

The text has been edited accordingly (lines 186-189 inclusive of the reference).

P. 9 lines 201-203: "This analysis indicated that the complemented mutant produced significantly less AM than the wild type (Fig. 3d), showing that complementation is imperfect, and suggesting that expression of the lamH gene is altered when located distally." This suggests that there is reduced expression of lamH in the complement strain, but the complemented gene is in the pMV306 plasmid driven by the Hsp60 promoter and therefore would likely be overexpressed. Please provide a different potential explanation for the reduced AM levels, or use RT-qPCR to demonstrate that there is truly lower expression of the lamH gene.

We thank the reviewer for considering this point. The pMV306 vector lacks a promoter of its own; there is no Hsp60 in this context, and thus, the gene is not overexpressed. We cloned the gene with 300 bp of the upstream DNA into the multi-cloning site of pMV306 to catch any likely promoter elements. This was because, despite careful analysis, we could not detect any clear promoter elements, which is not uncommon for mycobacteria. Consequently, any expression of *lamH* in this context is from its native promoter and highly unlikely to be above the WT's. We have edited the text (lines 194-199) to be more explicit about the nature of this construct.

P. 9 line 209: We suggest tempering the claim that "increased abundance of LM/LAM triggers a membrane stress response."

We have edited the sentence to temper this claim (line 222-225).

P.10, line 224: consistent verb tense, change "are" to "were"

We have edited the sentence to reflect the Reviewer's suggestion (lines 241-242).

Line 225: "this provides evidence that levels of LM/LAM are tightly regulated" We suggest tempering the language by removing "tightly" unless there is specific evidence to support this qualification; the point is otherwise still significant, that there regulation in the balance of intermediates/products in this pathway at all.

We have removed the word 'tightly' (line 251-253).

P.10/11, all data regarding the analysis of exponential growth rates (lines 236-240, lines 246-252, figure 5b,d). These arguments are quite complicated, and indeed for the rescue experiments the observations become less consistent (partial rescue of lag time, but decrease in growth rate). More to the point, the changes (up to 20%) in growth rate, although statistically significant, are not clearly biologically meaningful. Overall, we see these as comparatively weak arguments that are not necessary for the main conclusion that LamH activity is required for efficient exit from lag-phase growth and recommend that they be removed.

In line with the reviewer's suggestion, we have removed this analysis.

P.12, line 270: Significant figures. Unless the authors can show that a 1.01-log reduction is reproducible, we recommend that they report a smaller number of significant figures (e.g., 0.4 and 1).

We have updated the text to reflect more appropriate significant figures (line 295).

More importantly, Figure 6 is reported as "data representative of two independent experiments with similar results". If this is the case, then how many replicates are represented in the figure to support the statistical analysis? What statistical analysis was performed?

We have modified the materials and method section to indicate that each independent experiment included three biological replicates. Multiple t-test included a Two-stage linear step-up procedure of Benjamini, Krieger and Yekutieli, with $Q = 1\%$. Each row was analysed individually without assuming a consistent standard deviation. Number of test performed: 4. (Figure 6)

P. 13 lines 301-311: The three hypotheses for LM and LAM release to the host do not make sense as distinct possibilities. The mention of LprG likely only accounts for transport across the inner membrane, unless the authors are invoking a hypothesis through which LM and LAM are somehow secreted bound to LprG. The mentions of vesicle production and the presence of host antibodies against capsular AM do not sufficiently explain release of LM and LAM. Moreover, AM is a distinct species from LM/LAM and its presence in the capsule does not explain how LM/LAM reach host compartments. Please clarify the ideas in this section.

We appreciate that this text could have been more clearly written. We want to be clear that we are not stating these are mutually exclusive. It is possible that LM and LAM exit the cell through several mechanisms. Line 324 has been edited to make this clearer.

However, we believe the Reviewer may have a different understanding of the literature than we do surrounding LprG. Two key papers on LprG suggest that LprG chaperones LAM to the outside of the cell^{8,9}. We believe the Reviewer may be mistaken about the proposed role of LprG when they suggest it is involved in transit across the inner membrane. LprG is a Sec-secreted lipoprotein and thus would only

be folded and able to bind LAM in the periplasm. The hypothesis put forward in these papers would suggest that it helps expose LAM on the cell's surface. Some of the data supporting this statement are that in $\Delta/lprG$ strains, more LAM is detected by antibodies at the cell surface. Those data have methodological limitations because the antibodies used in that study detect carbohydrate moieties and so do not specifically recognise LAM because they would cross-react to AM. A loss of LprG also does not completely obliterate surface 'LAM' detection. Nevertheless, there is enough data to warrant presenting it as a possibility. We would also argue that vesicle production may, in fact, be sufficient for LAM release/exposure to the host. Work from the Kiessling group and others has shown that under some conditions, mycobacteria can release large quantities of vesicles, which have also previously been shown to include LAM^{10,11}.

We agree that AM is a distinct species. However, much of the literature around LAM trafficking suffers from the same methodological limitations as discussed above. These principally use antibodies for the carbohydrate rather than lipid epitopes on the molecule and cannot distinguish between LAM and AM. Consequently, we cannot rule out the possibility that at least some of the phenotypes attributed to LAM are, in fact, driven by AM. This would be a major shift in thinking in tuberculosis research, requiring further study. Fortunately, we are now able to do this, given that we can generate strains that are devoid of detectable AM as well as produce highly purified AM.

Nevertheless, we have edited the text for increased clarity around LprG (line 328-331).

Also, within this section, according to the literature, LprG has the capacity to bind to a variety of diacyl/triacyl lipids including LM/LAM and triacylglycerides, so roles in the transport of both classes are not necessarily mutually exclusive. We suggest rewording.

We have edited for clarity (line 331).

Most crucially, regarding the key statement, "this work provides the first mechanistic basis for the generation of capsular AM", overlooks the fact that LamH does not appear to have a predicted canonical secretion signal (or any N-terminal features consistent with one, such as even a weak predicted TM helix). Its presumed location in the cytoplasm is then inconsistent with a role in generating AM from LM/LAM in the periplasm.

Please see our response to Reviewer 1 on this same point. Our data overwhelmingly support this role, and prior proteomics datasets localise the protein to the cytoplasmic membrane¹⁻⁴. We agree it is unusual that it achieves this role without a canonical signal sequence. However, there is a precedent in the literature for mycobacterial proteins secreted without an identifiable signal peptide or transmembrane helix⁶. The mechanism of LamH secretion/localisation will be examined in future studies by our research group.

Moreover, the model for LAM/LM synthesis shown in Fig. 1 and 3 is not fully consistent with commonly reported models. We recommend that the authors more

clearly depict the ambiguity in the identity and localization of the GTs that create AcPIM3/4 from AcPIM2. Indeed, most models depict AcPIM4 as the substrate of a presumed flippase that moves this species into the periplasm, consistent with the apparent bifurcation between further elaboration into AcPIM5/6 vs. LAM/LM, both by periplasmic GTs.

We have edited the figure to reflect these comments and thank the reviewer for the helpful suggestion.

If we instead presume that AcPIM4 is first made in the cytoplasm, then cytoplasmic LamH could be involved in regulating the relative pools of AcPIM1 and AcPIM4, as a means of regulating the metabolic flux into downstream pathways. Admittedly, while this is consistent with the increase in LAM/LM in the lamH mutant, it does not easily explain the concomitant reduction in AM. Nevertheless, the authors should address the issue of enzyme localization in reconciling their model for the physiological role of LamH..

We believe this concern is satisfactorily addressed, given the published data localising LamH to the cytoplasmic membrane. However, precisely how and when it is exposed to the periplasm remains ambiguous and is the subject of further study in our laboratory.

METHODS:

P. 15: Please include methods for bioinformatic protein analyses.
These have been included (lines 366-376).

P. 33-34: Please include the enzyme concentration and the solvent system also in the figure legends throughout the manuscript. Also, please report the number of biological replicates performed even if only showing representative data.

We have reviewed the manuscript and addressed this where appropriate.

Line 347/349: Reported culture temperatures are 37C for yeast and 30C for M. smegmatis– is this correct? It seems the reverse of the typical growth temps for these microbes.

This is indeed a typographical error. We have corrected it on lines 385 and 387.

Line 364: Please clarify this section and heading to reflect that this mutagenesis is to generate a lamH KO. Please also include the MSMEG gene identifier here and the oligonucleotide sequence (there is no need for a supplemental table for a single oligo).

We have amended the text accordingly (line 402).

P. 16, line 378: “nearby genes” Please be more specific about which genes and also report the primer sequences used. More generally, we recommend that in reporting this analysis, the authors describe the context for lamH. Is it a single-gene operon? Do the surrounding genes have known or predicted functions that could be related?

Did the authors also do a synteny analysis and see any conservation of the gene context?

The genetic context for *lamH* did form some of our initial analysis of this gene; however, we did not feel it was sufficiently informative to include in this manuscript. A Webflags analysis of the locus reflects limited synteny, restricted to closely related species. The genes conserved in this area in the *M. tuberculosis* complex appear to have no functional relevance to *lamH*. In *M. tuberculosis* complex species, at the 5' end of *lamH* is a toxin-antitoxin pair (Rv0366c and Rv0367c) encoded in the same orientation. Consequently, while it is unlikely that interruption of *lamH* would impact their expression, we considered this in our experimental design. Our complementation studies, especially with the catalytic null variant of the gene and with chemical rescue, allow us to be confident that at least the proximal cause of the observed phenotypes is loss of LamH catalysis and not polar effects on this 5' pairing. Furthermore, our complementation construct does not include a constitutive promoter but instead uses the 300 bp 5' to Rv0365. This region does not include any predicted open reading frames. Complementation from this plasmid supports the conclusion that the promoter elements for *lamH* are contained within the region immediately 5' to *lamH* rather than *lamH* being encoded in an operon with Rv0366c and Rv0367c. We do not see the same locus arrangement in *M. smegmatis* with no conserved genes within 6 genes upstream or downstream of MSMEG_0740. Despite this, the $\Delta lamH$ *M. smegmatis* strain phenocopies our *M. bovis* BCG mutant, further strengthening our conclusions. The regulation of *lamH* is a topic we will follow up on in future studies. Finally, the 'and nearby genes' statement was included in error and has been removed.

P. 20, line 501: Especially given that this is not necessarily a widely used protocol, please provide the resuspension volume and provide a more reproducible direction than "gently vortexed" – e.g., vortexed at the lowest setting?

The text has been amended (lined 560-566).

P.21, line 522: please be more specific regarding the "portable spectrophotometer" if this is somehow critical to how the measurements were made. Else, monitoring of OD600 is considered standard and this vague specification should be removed for clarity.

We included this only because the work is conducted at BSL2 in a biosafety cabinet. This detail might be relevant to people repeating this work. Nevertheless, this text has been edited to remove the reference to the instrument.

P. 21, line 530: Please provide the methods for the preparation of the "acetone-precipitated proteome samples".

This has been added (line 587-596).

FIGURES:

Figure 1. We appreciate the authors' effort to depict the glycosidic bond connectivities, but the similar colors used for both sugars and the very short bonds

makes these differences indiscernible. Also, the caption states “Grey text indicates glycolipid species”, but grey text appears instead to name the GTs? Please amend for clarity/correctness.

While we appreciate that the green for both mannose and arabinose is similar, we have followed the accepted symbolic nomenclature for glycans standard used broadly in the glycobiology field (see: https://www.ncbi.nlm.nih.gov/glycans/snfg1_5.html). The bond depictions leave us with more room for artistic licence however, and so we have edited the figure to make each linkage type more distinct by increasing line widths. We have corrected the legend.

Figure 2. As noted above, please report the number of independent experiments that these data represent, for each experiment.

These were all conducted with 3 biological replicates. We have included this in the legend.

Figure 3. The caption and numbering appears to be off– i.e., a/b used to be combined LAM/LM analysis, but now are separate. Also, “Peaks were identified by digestion with appropriate enzymes” is vague and these methods do not appear to be provided in the methods section. Please amend.

We have edited this legend and added the digestion details to the methods (Figure 3, line 579).

Figure 5. Please clarify the legends vs. the caption. The legends between a), which refers to strains, and b), which refers to type of spent medium, is confusing. Also, the legend states “+AM”, consistent with this being a LamH-digested preparation of LAM, but the caption says “0.5 mg/mL LAM”. Please amend.

We have edited this legend accordingly. The legend now reads 0.5 mg/mL AM.

Figure 6. Please see comments above regarding # of independent replicates shown and subjected to statistical analysis, given that the caption states that data are representative of 2 experiments.

We have edited this legend accordingly.

1. Souza, G. A. D., Leversen, N. A., Målen, H. & Wiker, H. G. Bacterial proteins with cleaved or uncleaved signal peptides of the general secretory pathway. *J. Proteom.* **75**, 502–510 (2011).
2. Målen, H., Pathak, S., Søfteland, T., Souza, G. A. de & Wiker, H. G. Definition of novel cell envelope associated proteins in Triton X-114 extracts of *Mycobacterium tuberculosis* H37Rv. *Bmc Microbiol* **10**, 132–132 (2010).
3. Xiong, Y., Chalmers, M. J., Gao, F. P., Cross, T. A. & Marshall, A. G. Identification of *Mycobacterium tuberculosis* H37Rv Integral Membrane Proteins by One-

Dimensional Gel Electrophoresis and Liquid Chromatography Electrospray Ionization Tandem Mass Spectrometry. *J. Proteome Res.* **4**, 855–861 (2005).

4. Mawuenyega, K. G. *et al.* Mycobacterium tuberculosis Functional Network Analysis by Global Subcellular Protein Profiling. *Mol Biol Cell* **16**, 396–404 (2004).

5. Hermann, C., Giddey, A. D., Nel, A. J. M., Soares, N. C. & Blackburn, J. M. Cell wall enrichment unveils proteomic changes in the cell wall during treatment of Mycobacterium smegmatis with sub-lethal concentrations of rifampicin. *J. Proteom.* **191**, 166–179 (2019).

6. Perkowski, E. F. *et al.* The EXIT Strategy: an Approach for Identifying Bacterial Proteins Exported during Host Infection. *Mbio* **8**, e00333-17 (2017).

7. Nguyen, P. P., Kado, T., Prithviraj, M., Siegrist, M. S. & Morita, Y. S. Inositol acylation of phosphatidylinositol mannosides: a rapid mass response to membrane fluidization in mycobacteria. *J. Lipid Res.* **63**, 100262 (2022).

8. Gaur, R. L. *et al.* LprG-Mediated Surface Expression of Lipoarabinomannan Is Essential for Virulence of Mycobacterium tuberculosis. *Plos Pathog* **10**, e1004376 (2014).

9. Shukla, S. *et al.* Mycobacterium tuberculosis Lipoprotein LprG Binds Lipoarabinomannan and Determines Its Cell Envelope Localization to Control Phagolysosomal Fusion. *PLoS Pathog.* **10**, e1004471 (2014).

10. Wuo, M. G. *et al.* Antibiotic action revealed by real-time imaging of the mycobacterial membrane. *bioRxiv* 2022.01.07.475452 (2022)
doi:10.1101/2022.01.07.475452.

11. Athman, J. J. *et al.* Bacterial Membrane Vesicles Mediate the Release of Mycobacterium tuberculosis Lipoglycans and Lipoproteins from Infected Macrophages. *J. Immunol.* **195**, 1044–1053 (2015).

REVIEWERS' COMMENTS

Reviewer #1 (Remarks to the Author):

No further comments. Authors addressed all concerns raised in the previous manuscript. It is an excellent revision.

Reviewer #3 (Remarks to the Author):

We thank the authors for their responsiveness to the comments. The revisions significantly clarified and strengthened the manuscript. We have only a few additional minor suggestions and otherwise find the manuscript appropriate and sound.

Regarding the subcellular location of Rv0365c, we suggest that the authors add one additional comment from the rebuttal to the discussion to further strengthen the idea that Rv0365c could be in the cell wall as is most consistent with the authors' proposed function, namely adding: "There is a precedent in the literature for mycobacterial proteins secreted without an identifiable signal peptide or transmembrane helix" along with its citation.

Very minor:

Lines 81-64: "Release of these molecules may happen passively, as is the case during cell wall degradation because of division and wall remodelling, or actively, as the bacteria secrete effectors into the host in a manner resembling the release of peptidoglycan fragments in other bacteria." It is not entirely clear what the distinction between "passive" and "active" is in this context, as cell wall degradation is also the result of enzymatic activity. Please reword for clarity.

Line 400: "incubated in static" - Please clarify what is meant by "static" with a brief comment about why this condition was chosen, if it is not in fact standard aerating culture.

Reviewer #4 (Remarks to the Author):

The Authors have fully addressed my comments in their revision and I have no further concerns.

Response to reviewers:

Reviewer #1 (Remarks to the Author):

No further comments. Authors addressed all concerns raised in the previous manuscript. It is an excellent revision.

We thank the reviewer for their evaluation of the manuscript.

Reviewer #3 (Remarks to the Author):

We thank the authors for their responsiveness to the comments. The revisions significantly clarified and strengthened the manuscript. We have only a few additional minor suggestions and otherwise find the manuscript appropriate and sound.

We thank the reviewer for their evaluation of the manuscript.

Regarding the subcellular location of Rv0365c, we suggest that the authors add one additional comment from the rebuttal to the discussion to further strengthen the idea that Rv0365c could be in the cell wall as is most consistent with the authors' proposed function, namely adding: "There is a precedent in the literature for mycobacterial proteins secreted without an identifiable signal peptide or transmembrane helix" along with its citation.

We have added the suggested edit, at lines 116-118.

Very minor:

Lines 81-64: "Release of these molecules may happen passively, as is the case during cell wall degradation because of division and wall remodelling, or actively, as the bacteria secrete effectors into the host in a manner resembling the release of peptidoglycan fragments in other bacteria." It is not entirely clear what the distinction between "passive" and "active" is in this context, as cell wall degradation is also the result of enzymatic activity. Please reword for clarity.

This text has been edited for clarity. Lines 79-82.

Line 400: "incubated in static" - Please clarify what is meant by "static" with a brief comment about why this condition was chosen, if it is not in fact standard aerating culture.

We disagree that this term needs clarification. Static has a well-defined meaning (not moving) and indicates that the culture is not aerating. These are standard growth conditions for *M. bovis* BCG, used in many laboratories.

Reviewer #4 (Remarks to the Author):

The Authors have fully addressed my comments in their revision and I have no further concerns.

We thank the reviewer for their evaluation of the manuscript.